# The NP protein of Newcastle disease virus dictates its oncolytic activity by regulating viral mRNA translation efficiency

Tianxing Liao[1][◉], Yu Chen[1][◉], Lili Guo[1], Shanshan Zhu[1], Tiansong Zhan[1], Xiaolong Lu[1], Haixu Xu[1], Zenglei Hu[1,2,3,4], Jiao Hu[1,2], Min Gu[1,3], Xiaowen Liu[1,2], Xiaoquan Wang[1], Shunlin Hu[1,2,3]*, Xiufan Liu[1,2,3,4]*

1 Animal Infectious Disease Laboratory, College of Veterinary Medicine, Yangzhou University, Yangzhou, China, 2 Jiangsu Co-innovation Center for Prevention and Control of Important Animal Infections Diseases and Zoonoses, Yangzhou University, Yangzhou, China, 3 Jiangsu Key Laboratory of Zoonosis, Yangzhou University, Yangzhou, China, 4 Joint International Research Laboratory of Agriculture and Agri-Product Safety, Ministry of Education of China, Yangzhou University, Yangzhou, China

◉ These authors contributed equally to this work.
* slhu@yzu.edu.cn (SH); xfliu@yzu.edu.cn (XL)

**Data Availability Statement:** The high-throughput data has been uploaded to the GEO database (GSE233617) (https://www.ncbi.nlm.nih.gov/geo/query/acc.cgi?acc=GSE233617).The raw data from

## Abstract

Newcastle disease virus (NDV) has been extensively studied as a promising oncolytic virus for killing tumor cells in vitro and in vivo in clinical trials. However, the viral components that regulate the oncolytic activity of NDV remain incompletely understood. In this study, we systematically compared the replication ability of different NDV genotypes in various tumor cells and identified NP protein determines the oncolytic activity of NDV. On the one hand, NDV strains with phenylalanine (F) at the 450th amino acid position of the NP protein (450th-F-NP) exhibit a loss of oncolytic activity. This phenotype is predominantly associated with genotype VII NDVs. In contrast, the NP protein with a leucine amino acid at this site in other genotypes (450th-L-NP) can facilitate the loading of viral mRNA onto ribosomes more effectively than 450th-F-NP. On the other hand, the NP protein from NDV strains that exhibit strong oncogenicity interacts with eIF4A1 within its 366–489 amino acid region, leading to the inhibition of cellular mRNA translation with a complex 5' UTR structure. Our study provide mechanistic insights into how highly oncolytic NDV strains selectively promote the translation of viral mRNA and will also facilitate the screening of oncolytic strains for oncolytic therapy.

## Author summary

The oncolytic potential of NDV has gained significant attention in the context of clinical trials. Therefore, it is crucial to systematically compare the oncolytic activities of different NDV subtypes and understand their underlying mechanisms. Our previous investigations revealed that genotype VII NDVs, predominant during the fourth epidemic, exhibit heightened avian pathogenicity compared to other subtypes. Interestingly, in tumor cells, NDV strains of varying virulence and genotypes possess some oncolytic capacity.

mass spectrometry sequencing has been uploaded to the iProX database (PXD044165) (https://www.iprox.cn//page/project.html?id=IPX0006791000). Other relevant data are within the paper and its Supporting Information files.

**Funding:** This work was supported by the National Natural Science Foundation of China (32202767 to Y.C), the State Key Laboratory of Veterinary Biotechnology Foundation (SKLVBF202205 to Y.C), the Natural Science Foundation of Jiangsu Province (BK20210077 to H.X), the Priority Academic Program Development of Jiangsu Higher Education Institutions (PAPD to X.L), and the National Natural Science Foundation of China (31702243 to Z.H). The funders had no role in study design, data collection and analysis, decision to publish, or preparation of the manuscript.

**Competing interests:** The authors have declared that no competing interests exist.

However, in several genotype VII strains, the NP protein fails to facilitate the efficient loading of viral mRNA onto ribosomes. This is primarily attributed to the presence of phenylalanine at the 450th amino acid position of the NP protein. Additionally, it loses its ability to inhibit the eIF4A1-relative cellular mRNA translation through interaction with eIF4A1, mainly because of differences in its non-conserved region (366-489aa) of NP protein compared to other genotypes. These significantly hinder its ability to establish effective infections in tumor cells. Our study provides valuable insights for oncolytic NDV strain screening and enhances understanding of the functional role of NP protein.

## Introduction

Newcastle disease virus is a member of *Avian orthoavulavirus 1* (AOAV-1) with a genome length of approximately 15.2 kb, encoding six viral proteins: nucleocapsid protein (NP), phosphoprotein (P), matrix protein (M), fusion protein (F), hemagglutinin-neuraminidase (HN) and large protein (L). Depending on the intensity of the illness it causes in birds, NDV is divided into three pathotypes: lentogenic (avirulent), mesogenic (middle), or velogenic (virulent) [1]. As a promising oncolytic virus, NDV can infect a wide range of cells and has a replication capacity 10,000 times greater on tumor cell lines than on normal cells [2]. With the development of lentogenic strains HUJ, Ulster, La Sota, Hitchner B1, and V40-UPM as oncolytic vaccines [3], lentogenic, mesogenic, or velogenic strains have all been shown to be tumorolytic. NDV exerts a direct anti-tumor effect by affecting the activity of infected cells and inducing their death. Additionally, NDV infection causes an inflammatory environment in the tumor microenvironment, which can directly activate NK cells, monocytes, macrophages, and dendritic cells and promote the recruitment of immune cells to exert an indirect anti-tumor effect [4]. Thus, the anti-tumor efficacy of NDV is mainly dependent on its ability to infect tumor cells. While lentogenic and mesogenic NDV strains of genotypes II and III are commonly used in clinical trials, there needs to be a more systematic comparison of the infectivity of different NDV genotypes with varying virulence on different cell lines.

The NP protein is the most abundant viral protein in NDV particles, with an open reading frame 1470 bp in length, encoding 489 amino acids with two major regions: a highly conserved structured N-terminal region forming spherical vesicles (Ncore) and a C-terminal region extending from the main N-terminal body exposed on the surface of the assembled nucleocapsid (Ntail) [5–8]. Ncore contains all the components required for NP self-assembly and RNA binding to form the NP-RNA complex, while the Ntail is primarily responsible for the interaction of the NP-RNA complex with P proteins [9]. Thus, the binding of the NP-RNA template to the RdRp complex occurs primarily through the Ntail of the NP [10], which in turn affects the activity of the viral polymerase. Since the N protein of Measles virus (MeV) is involved in the repression of cellular translation [11], and the NP protein of NDV is involved in the regulation of mRNA translation initiation by interacting with eIF4E [12], the NP protein of NDV is thought to have an important role in the regulation of viral mRNA translation.

Translation in eukaryotes always begins with the following stages: eIF2 guides the ternary complex to the 40S ribosomal subunit, forming the 43S pre-initiation complex that contains eIF3. The eIF3 promotes the binding of the 43S pre-initiation complex to mRNA through its interaction with the cap-binding complex eIF4F, which is already assembled around the mRNA 5' m$^7$G cap [13,14]. eIF4F is a protein complex consisting of eIF4A1, eIF4E, and eIF4G [15–17]. Translational control usually occurs at the translation initiation step, in which ribosomes are recruited to the 5' cap of the mRNA. First, eIF4E, as part of the eIF4F complex,

promotes the recruitment of the 40S ribosomal subunit by interacting with the 5′ terminus of the mRNA [18]. eIF4G plays a scaffolding role by interacting with both eIF4E and eIF4A1. Notably, ribosomes have a weak capacity to unwind mRNA secondary structures, while eIF4A1 has the ability to unwind stable secondary structures in the 5′-UTR during scanning. Thus, eIF4F does not control the translation of all mRNAs, but rather a subset that possess specific secondary structures in their 5' UTRs [19]. The translation of viral mRNA entirely depends on the host translation system, with translation initiation being the major rate-limiting step. Influenza virus RNA polymerase and NS1 interact with eIF4G to recruit eIF4F for the translation of viral mRNAs [20]. MeV directly recruits eIF3-p40 via the N protein [11], the M protein of Rabies virus is involved in the regulation of translation initiation by binding to eIF3h, and the M protein of VSV facilitates the formation of the initiation complex by inhibiting the phosphorylation of 4EBP1 [21]. However, how the virus recruits ribosomes via the altered translation initiation complex is still unknown. Modulating the modification, activity, and expression of the translation initiation complex is also a strategy viruses use to regulate host translation. For instance, influenza virus, VSV, or adenovirus can reduce the expression of phosphorylated eIF4E [22–24]. In addition to translation initiation, regulation of the elongation and termination processes also plays an important role. Unfortunately, research on the NDV-regulated translation system in cells is limited.

Our previous studies have demonstrated that genotype VII NDV causes more severe damage to avian immune tissues and has greater infectivity on avian immune cells in vitro [25–27]. In this study, we compared the infectivity of various subtypes of NDV strains with different virulence on various tumor cells and found that several genotype VII NDVs are not oncolytic. Using a reverse genetics approach, we constructed multiple recombinant viruses and identified the functional gene responsible for the difference as NP. Our findings indicate that the 450th amino acid residue of the NP protein plays a crucial role in promoting the loading of viral mRNA onto ribosomes. Additionally, the NP protein of the highly infectious Herts/33 strain interacts with eIF4A1 through its 366-489aa region, inhibiting the translation of host mRNA that is dependent on eIF4A1. This study provides important insights into the function of the NP protein, as well as the screening of NDV tumor lysis vaccine strains and the construction of recombinant vaccine strains.

## Materials and methods

### Ethics statements

All NDV strains investigated in this study were exclusively employed to explore their oncolytic potential, aiming to contribute to cancer treatment. The mutations or substitutions in all recombinant viruses studied in this research were naturally occurring in the environment. All experiments involving NDV were executed in the animal biosafety level 3 facility (CNAS registration No. CNAS BL0015) at Yangzhou University in strict accordance with the recommendations of the institutional biosafety manual and supervised by the Institutional Biosafety Committee of Yangzhou University.

### Cell culture and virus

The human epithelial carcinoma cell line HeLa, human non-small cell lung cancer cell line A549, human colon cancer cell line HT29, human colorectal adenocarcinoma cell line HCT116 and the DF1 Chicken Embryo Fibroblast cell line were purchased from the American Type Culture Collection (ATCC) and maintained in Dulbecco's modified Eagle's medium (DMEM) supplemented with 10% fetal calf serum (Thermo Fisher Scientific, Waltham, MA, USA), in an incubator at 37°C containing 6% $CO_2$. The undifferentiated thyroid cancer

SW1736 cell line was generously provided by Professor Xiulong Xu [28]. The primary Chicken Embryo Fibroblast CEF cell line was prepared in our laboratory. NDV strain Herts/33 (AY741404.1), ZJ1 (AF431744.3), La Sota (KC844235.1), Mukteswar (EF201805.1), JS-7122 (FJ430159.1, also known as JS705Ch), Kuwait (MK978147.1) and F48E8 (FJ436302.1) was obtained from the China Institute of Veterinary Drug Control (Beijing, China). $I_4$ (also known as JS-5-05-Go, JN631747.1), YN1200 (KT760569.1), JS1816 (MH105251.1), JS-01-08-Ch (JQ013853.1), JS-1-06-Ch (EU044804.1), JS-1-97-Go (AY847294.1), JS-3-05-Ch (JN618349.1), JS-08-11-Go (KJ184633.1), JS-09-12-Ch (KJ184625.1), JS-10-12-Ch (KJ184580.1), JS-12-12-Ch (KJ184582.1), JS-21-13-Ch (KP064023.1) and JS-30-13-Ch (KP064026.1) were isolated by our laboratory.

## Antibodies and reagents

Cycloheximide (CHX, 239764) was purchased from Sigma-Aldrich and used at 100 μg/ml. Enhanced Cell Counting Kit-8 (#C0042) and Annexin V-FITC Cell apoptosis detection kit (#C1062S) were purchased from Beyotime (China). PrimeScriptRT reagent Kit with gDNA Eraser (RR047B), TB Green Premix Ex Taq II (RR820W) and *Premix Ex Taq*(Probe qPCR) (RR390W) was purchased from TAKARA. Human interferon α (IFN-α) ELISA kit (HM10251) and Human interferon β (IFN-β) ELISA kit (HM10099) were purchased from Bioswamp (China, Wuhan). Mouse monoclonal antibody against NDV Hemagglutinin-Neuraminidase protein (HN protein) was purchased from Santa Cruz Biotechnology (Dallas, TX, USA). Rabbit polyclonal antibody against NDV nucleocapsid protein (NP protein) was prepared in our laboratory. Puromycin and mouse monoclonal anti-puromycin 12D10 (MABE343) were purchased from Merck Millipore (Billerica, MA, USA). Rabbit monoclonal anti-His antibody (RM146), mouse monoclonal anti-HA-tag antibody (05–904) and anti-β-actin antibody (A1978), Horseradish peroxidase (HRP)-conjugated goat anti-rabbit (AP510P) secondary antibody, and horseradish peroxidase (HRP)-conjugated rabbit anti-mouse (AP160P) secondary antibody was purchased from Sigma-Aldrich (St. Louis, MO, USA). The secondary antibody goat anti-mouse IgG(H+L)-FITC (#1036–02, Southern Biotech) and goat anti-rabbit IgG(H+L)-AF647 (#4010–31, Southern Biotech) was purchased from Southern Biotech. Phospho-IRF-3 (Ser386) rabbit mAb (#37829) was purchased from Cell Signaling Technology. MAVS rabbit pAb (A5764), phospho-STAT1-Y701 rabbit mAb (AP0054), STAT1 rabbit mAb (A19563), phospho-STAT2-Y690 rabbit pAb (AP0284), STAT2 rabbit pAb (A14995), PKR/EIF2AK2 rabbit mAb (A19545), phospho-PKR-T446 rabbit mAb (AP1134), eIF2α rabbit pAb (A0764), phospho-eIF2α-S51 rabbit pAb (AP0342) and IRF3 rabbit mAb (3661) were purchased from Southern ABclonal. LY294002 (HY-10108), SC79 (HY-18749), and Rocaglamide (Roc-A, HY-19356) were obtained from MedChemExpress (MCE). NP-40 Lysis Buffer (P0013F), Protein A+G Agarose (P2055-2ml), rabbit IgG (A7016), Myc rabbit polyclonal antibody (AF6513), cyclin B1 rabbit monoclonal antibody (AF1606), cyclin A2 rabbit monoclonal antibody (AF2524), Akt mouse monoclonal antibody (AF0045), phospho-Akt(Ser473) antibody (Rabbit Polyclonal) (AA329), and phospho-P70 S6K (Thr389) rabbit polyclonal antibody (AF5899) were purchased from Beyotime. PDCD4 rabbit monoclonal antibody (A9068) was obtained from ABclonal. pBiFC-VN173 (GS-1161) and pBiFC-VC155 (GS-1162) were purchased from AKYBio (China).

## Cell infection and viral growth kinetics

Cells were infected with NDV at a multiplicity of infection (MOI) of 0.1, 1, or 10, respectively. After one hour of adsorption at 37˚C, the unattached virus was removed, and cells were washed three times with PBS and incubated in a maintenance medium at 37˚C. Virus titers

were determined as half tissue infection ($TCID_{50}$) using the Reed and Muench method (1938) at 72 hours post-infection (hpi), as previously described [29]. To determine the $TCID_{50}$ value of La Sota and its recombinant virus, cells were cultured in a medium supplemented with 5μg/ml of N-tosyl-L-phenylalanyl chloromethyl ketone [TPCK]-treated trypsin (Sigma-Aldrich).

To measure the growth kinetics of two NDV strains, Herts/33 and $I_4$ in HeLa cells under multi-cycle growth conditions, cells were seeded onto 12-well plates and infected with NDVs at an MOI of 0.1, 1, or 10. Culture supernatants were collected every 12 hours until 72 hpi, and virus titers in the supernatant were determined by the limited dilution method and expressed as $TCID_{50}$ value. All experiments were repeated three times.

## Cell viability assay

The impact of rHerts/33 and $rI_4$ on cell viability in the HeLa cell line was assessed using the Enhanced Cell Counting Kit-8 (#C0042, Beyotime, China), following the manufacturer's instructions. In brief, $1.5 \times 10^3$ cells per well were seeded in 96-well plates and incubated for 12 hours. And then they were infected with NDVs at different MOIs (0.1, 1, 10). The absorbance of each well was measured and expressed as a percentage relative to the control. Each experiment was repeated three times.

## Construction of Plasmid

Recombinant viruses were generated by exchanging a single NP, P, M, F, HN, or L gene or a combination thereof between $I_4$ (same as JS-5-05, GenBank accession number: JN631747.1) and Herts/33 (GenBank accession number: AY741404.1). Restriction enzymes including SgrAI (15735), AgeI (2879), Bstz17I (4705), ApaI (2289), PacI (2899), and SpeI (8094) were used. The recombinant virus rH-NPPL$^I$ and rI-NPPL$^H$, simultaneously replacing the NP, P, and L genes, was constructed by Yan Kai [27]. Detailed construction strategies and primers of other recombinant viruses are presented in S1 Table. All parental and recombinant viruses were rescued through plasmid reconstitution to prevent contamination. Therefore, we prefixed the letters "r" to their names to designate them.

Point mutations were introduced by PCR, resulting in the mutations rI-NP$_{F450L}$, rI-NP$_{P464S}$, rI-NP$_{S479P}$, rH-NP$_{L450F}$, rH-NP$_{S464P}$, rH-NP$_{P479S}$, rK-NP$_{L450F}$, and rL-NP$_{R450F}$. Specifically, rI-NP$_{F450L}$ indicates a mutation from phenylalanine to leucine at 450aa of the $I_4$ NP protein, rI-NP$_{P464S}$ indicates a mutation from proline to serine at position 464aa of the $I_4$ NP protein, rI-NP$_{S479P}$ indicates a mutation from serine to proline at position 479aa of $I_4$ NP protein, rH-NP$_{L450F}$ indicates a mutation from leucine to phenylalanine at 450aa of Herts/33 NP protein, rH-NP$_{S464P}$ indicates a mutation from serine to proline at 464aa of Herts/33 NP protein, rH-NP$_{P479S}$ indicates a mutation from proline to serine at 479aa of Herts/33 NP protein, rK-NP$_{L450F}$ indicates a mutation from leucine to phenylalanine at 450aa of Kuwait NP protein, and rL-NP$_{R450F}$ indicates a mutation from leucine to phenylalanine at 450aa of La Sota NP protein. The sequences corresponding to all primers are listed in S1 Table.

The NP, P, and L genes from $I_4$ and Herts/33 were PCR amplified from cDNAs and cloned into the pCI-neo vector to generate helper plasmids, pCI-INP/ pCI-HNP, pCI-IP/ pCI-HP, and pCI-IL/ pCI-HL for the NP, P, and L proteins, respectively. NDV small genomic plasmids containing the GFP gene driven by the T7 promoter were also constructed following previous methods [30]. They are named TVT-IGFP and TVT-HGFP, respectively, and contain the leading and trailing sequences of $I_4$ and Herts/33. Mutations at amino acid 450 of the NP of $I_4$ and Herts/33 were introduced by PCR and named pCI-I450 and pCI-H450, respectively. Detailed construction strategies are presented in S2 Table.

To overexpress NDV NP truncated fragments and the eIF4A1 protein, the NP cDNA and its fragments were cloned into the vector pCAGGS to generate pCAGGS-HNP-His, pCAGGS-INP-His, pCAGGS-HNP$_{1-245}$-His, pCAGGS-HNP$_{245-366}$-His, pCAGGS-HNP$_{245-489}$-His, pCAGGS-HNP$_{366-489}$-His, and pCAGGS-HNP$_{122-366}$-His. His tag was introduced at the C-terminus of all NP and truncated NP plasmids. All plasmids were confirmed by sequencing. For transfection, cells were seeded in six-well plates and transfected with Lipofectamine 3000 when they reached 70% confluency following the manufacturer's instructions. The primers used in this part are listed in S3 Table.

To construct plasmids used in the BiFC experiments, we cloned the sequences of eIF4A1, HNP, and INP into pBiFC-VN173 and pBiFC-VC155 vectors. These constructs were named VC155-eIF4A1, VN173-HNP, and VN173-INP, respectively. The primer sequences for amplifying the respective fragments are listed in S4 Table. When constructing the plasmids used in the GST pull-down experiments, we selected the HindIII and EcoRI restriction enzyme cleavage sites within the pET-N-GST multicloning site as insertion sites. After amplifying each fragment with the respective primers, we performed homologous recombination by ligating them with the enzymatically cleaved and recovered fragment of pET-N-GST. This resulted in the construction of pET-GST-HNP and pET-GST-HNP prokaryotic expression vectors. The primer sequences for amplifying the respective fragments are listed in S6 Table.

## Rescue of virus from cDNA

BSR cells were infected with FPV-T7 for 1 hour, followed by co-transfection with full-length cDNA plasmids and helper plasmids expressing NP, P, and L, as described previously [31]. All parental and recombinant viruses used their respective helper plasmids and full-length cDNAs to avoid potential heterologous recombination in transfected cells. Culture supernatants were collected three days later, and viral stock solution was obtained by inoculating the supernatants into 9-11-day-old SPF chicken embryos. Total RNA was extracted from the NDV-positive allantoic fluid using Trizol reagent (Invitrogen). All parental strains and recombinant viruses were sequenced. No indeterminate mutations were detected. To rescue rLa Sota and its recombinant virus, transfected cells were maintained in a medium containing 5μg/ml N-tosyl-L-phenylalanyl chloromethyl ketone [TPCK]-treated trypsin (Sigma-Aldrich) [32].

All rescued viruses were identified using the intracerebral pathogenicity index (ICPI) test, as previously described [30]. Ten one-day-old SPF chicks were inoculated via the intracerebral route with 0.05 ml of a 1:10 dilution of fresh allantoic fluid infected with NDVs. Birds were monitored for clinical signs and mortality every 24 hours for 8 days. At each observation point, birds were scored as follows: normal (0), sick (1), or dead (2). The ICPI was calculated as the average score for each bird over the eight days. Pathotype definitions by the ICPI were as follows: virulent strains (1.50–2.00), moderately virulent strains (0.70–1.50), and avirulent strains (0.00–0.70).

## Phylogenetic analysis and sequence analysis of NP

We retrieved 890 full-length NDV genome sequences from NCBI and extracted the complete F gene sequences using MEGA. The nucleotide sequences were aligned and analyzed with MegAlign within the Lasergene package. Maximum likelihood (ML) phylogenetic trees were constructed using the F gene sequences (1–1662 nt) with the SYM model and 10000 bootstrap replicates in phylosuite (v1.2.2).

Furthermore, the full-length NP genes from the 890 strains were extracted using MEGA, and the corresponding protein sequences were compared.

## Flow cytometry

HeLa cells were seeded into 6-well plates and incubated for 12h before being infected with NDVs at a multiplicity of infection (MOI) of 10. Uninfected cells were used as a negative control. At 24 hours post-infection (hpi), cells were collected, washed with PBS, and resuspended and stained with the Annexin V-FITC Cell apoptosis detection kit (#C1062S) according to the manufacturer's instructions. Flow cytometry analysis was performed using a BD Biosciences instrument, and the data were analyzed using FlowJo software. All experiments were performed in triplicate.

For the virus binding assay, a previously described method was used [33]. Briefly, after seeding in 6-well plates, cells were incubated with NDV at 10 MOI for 1h at 4˚C. Subsequently, cells were collected and resuspended in PBS, incubated with HN antibody for 45 minutes at 4˚C, and then incubated with a FITC-labelled antibody for 45 minutes at 4˚C. The proportion of cells bound to NDVs was analyzed by flow cytometry. All experiments were performed in triplicate.

## qRT-PCR

Plasmid-transfected or virus-infected cells were harvested and treated with TRIzol reagent (Invitrogen) following the manufacturer's protocol. Total RNA was extracted and reverse transcribed using 2μg of RNA per sample, and then it was quantified by qRT-PCR as previously described [34,35]. *Premix Ex Taq* (Probe qPCR) (TaKaRa, Japan) was used for qRT-PCR experiments according to the manufacturer's instructions. The reaction volume was 25μL, containing 12.5μL of premix Ex Taq (2×) (Probe qPCR), 200nM of each primer, 1μL of probe, and 8.5μL of ddH$_2$O. The cycling conditions were 1 cycle at 94˚C for 30 s, followed by 40 cycles at 94˚C for 5 s, 60˚C for 30s. TB Green Premix Ex Taq II was used for relative quantitative PCR detection. The reaction mixture consisted of 12.5 μL of TB Green Premix Ex Taq II (2X), 10 μM of each primer, and ddH2O added to a total volume of 25 μL. The relative mRNA expressions were normalized to β-actin within each sample, and fold changes were calculated using the $2^{-\triangle\triangle Ct}$ method for relative quantification. All experiments were performed in triplicate. All primer and probe sequences are listed in S5 Table.

## Minigenome assays

The small genome assay was performed as previously described [35]. Briefly, BSR-T7/5 cells were transfected with TVT-IGFP/ TVT-HGFP, pCI-INP/ pCI-HNP, pCI-IP/ pCI-HNP, and pCI-IL/ pCI-HNP in 6-well plates using Lipofectamine 3000 (L3000015, Invitrogen). As a negative control, TVT-IGFP/ TVT-HGFP was replaced with the TVT empty vector to normalize the total amount of transfected DNA in the small genome system. Cells were collected at 24 post-transfection for real-time PCR and Western blotting analysis to detect small genome-specific RNA and protein levels. All experiments were repeated at least three times.

## Indirect immunofluorescence

To visualize cell infection, HeLa cells were seeded onto 24-well plates and infected with NDV at 0.1, 1, or 10MOI. At 24 and 48 hpi, cells were fixed with 4% paraformaldehyde, permeabilized with 0.25% Triton X-100, and blocked with 5% bovine serum albumin. Cells were then incubated with MAb-HN overnight at 4˚C. The secondary antibody Goat Anti-Mouse IgG(H+L)-FITC (#1036–02, Southern Biotech) was incubated for 1 hour at room temperature. Nuclei were stained with hoechst (#14533, merck, USA). To visualize the fluorescence of Venus protein in the Bifc experiment, HeLa cells were seeded onto 24-well plates and

transfected individually or co-transfected with the respective plasmids. After 18 hours, the cells were fixed using 4% paraformaldehyde, permeabilized with 0.25% Triton X-100, and nuclei were stained with Hoechst.

Fluorescence images were captured using a Leica TCS SP8 fluorescence microscope (Leica Microsystems GmbH, Wetzlar, Germany). Then they were analyzed and merged using Adobe Photoshop 2020 software (Adobe Systems, San Jose, CA, USA). All experiments were repeated at least three times.

## Ribopuromycylation method and confocal microscopy

The ribopuromycylation method (RPM) was performed as previously described [36]. Briefly, slides were placed into 24-well plates, and cells were seeded onto the slides and allowed to grow for 18 hours. The cells were then transfected with pCAGGS-HNP-His, pCAGGS-IN-P-His, or control empty vector, respectively, and incubated with emetine (208 M) for 15 minutes at 37˚C to terminate ribosome extension on the mRNA strand, 24 hours after transfection. Cells were washed with Dulbecco's modified Eagle's medium (DMEM) supplemented with 7.5% fetal bovine serum (FBS) (Atlanta Biologicals) and then incubated for 5 minutes at 37˚C in DMEM-FBS supplemented with 182M puromycin (PMY) and 208M emetine. After washing twice with cold PBS, cells were extracted and fixed for 20 minutes on ice in polysome buffer (50mM Tris-HCl [pH 7.5], 5mM $MgCl_2$, 25mM KCl, 355μM cycloheximide, EDTA-free protease inhibitor, 10 U/ml RNase inhibitor) supplemented with 200 mM NaCl, 0.1% Triton X-100, and 3% paraformaldehyde (PFA). Then cells were incubated with anti-His and anti-puromycin 12D10 antibodies overnight at 4˚C. The secondary antibody Goat Anti-Mouse IgG(H+L)-FITC (#1036–02, Southern Biotech) and Donkey Anti-Rabbit IgG(H+L)-AF555 (#6441–32, Southern Biotech) were incubated for 1 hour at room temperature. Nuclei were stained with hoechst (#14533, merck, USA). To investigate the co-localization of eIF4A1 with the NP protein, pCAGGS-eIF4A1-HA was transfected either individually or co-transfected with pCAGGS-HNP-His or pCAGGS-INP-His. After 18 hours, the cells were blocked for one hour and then incubated overnight at 4˚C with the mouse monoclonal anti-HA antibody and rabbit polyclonal anti-NP protein antibody.The secondary antibody Goat Anti-Mouse IgG(H+L)-FITC (#1036–02, Southern Biotech) and Donkey Anti-Rabbit IgG(H+L)-AF555 (#6441–32, Southern Biotech) were incubated for 1 hour at room temperature. Nuclei were stained with hoechst.

## Western blot

The NDV-infected cells and minigenome-transfected HeLa cells were washed with PBS and lysed with lysis buffer at 4˚C. The lysates were cleared by centrifugation at 12,000 × g for 15min. The resulting supernatant was subjected to SDS-PAGE under reducing conditions and transferred onto a PVDF membrane (Bio-rad). The membrane was blocked with a skimmed milk solution containing 0.1% Tween 20 in 5% tris-buffered saline for 1 hour at room temperature. It was then incubated overnight at 4˚C with the primary antibody, followed by incubation with HRP-conjugated secondary antibodies for 1h at room temperature. After washing, protein bands were detected using Supersignal West Pico (Thermo Fisher Scientific).

## RNA interference

The HeLa cells were transfected with siRNAs targeting RIG-I, PKR and eIF4A1 at a final concentration of 40nM using Lipofectamine 3000 (L3000015, Invitrogen) when the cells were at 60–70% confluency. At 36 hours post-transfection, the cells were infected with viruses at MOIs

of 1 and 10, respectively. The cells were collected at 24 hours post-infection and subjected to Western blot analysis. The sequences of siRNAs were listed in S7 Table.

## Polysome profile analysis

Based on previous studies [37,38], HeLa cells were infected with NDV for 12h or transfected with minigenome for 24h. Subsequently, cells were incubated with 100 μg/mL cycloheximide (CHX) at 37°C for 15 min, washed, and lysed for polysome fractionation via sucrose density gradient ultracentrifugation. Lysates were centrifuged at 12,000 × g for 15 min at 4°C, and the supernatant was then centrifuged at 40,000 × g for 3 h at 4°C on a linear sucrose gradient (10–50% buffer containing 20 mM Tris-Cl, pH 8.0, 140 mM KCl, 1.5 mM $MgCl_2$, 1 mM DTT, 1 mg/mL heparin). After centrifugation, each 1 mL fraction was collected, and UV absorbance at 254 nm was measured. To analyze protein in polysomes, total proteins from each sucrose gradient fraction were precipitated with trichloroacetic acid (TCA) and analyzed by Western blot. To examine the distribution of ribosomal RNA in the gradients, total RNA was extracted with phenol/chloroform, precipitated with ethanol, and resuspended in $dH_2O$ for quantitative RT-PCR analysis. Absolute quantitative analysis of NDV NP mRNA was performed using real-time PCR, as previously described. The primer sequences are listed in S5 Table.

## LC-MS/MS analysis

HeLa cells were transfected with pCAGGS-HNP-His or pCAGGS-INP-His. After 48 hours, the cells were lysed using NP-40, and 5 μg of rabbit IgG along with protein A+G beads were co-incubated with the cell lysates for 4 hours to remove non-specific binding. Subsequently, co-IP experiments were conducted using rabbit anti-NP polyclonal antibody, and the precipitated complexes were subjected to mass spectrometry analysis by Applied Protein Technology (APTBIO, China, Shanghai) 10 cm long, 75 μm inner diameter, 3μm resin) in buffer A (0.1% Formic acid) and separated with a linear gradient of buffer B (84% acetonitrile and 0.1% Formic acid) at a flow rate of 300 nl/min controlled by IntelliFlow technology. LC-MS/MS analysis was performed on a Q Exactive mass spectrometer (Thermo Scientific) that was coupled to Easy nLC (Proxeon Biosystems, now Thermo Fisher Scientific) for 60/120/240 min (determined by project proposal). The mass spectrometer was operated in positive ion mode. MS data was acquired using a data-dependent top10 method dynamically choosing the most abundant precursor ions from the survey scan (300–1800 m/z) for HCD fragmentation. Automatic gain control (AGC) target was set to 3e6, and maximum inject time to 10 ms. Dynamic exclusion duration was 40.0 s. Survey scans were acquired at a resolution of 70,000 at m/z 200 and resolution for HCD spectra was set to 17,500 at m/z 200, and isolation width was 2 m/z. Normalized collision energy was 30 eV and the underfill ratio, which specifies the minimum percentage of the target value likely to be reached at maximum fill time, was defined as 0.1%. The instrument was run with peptide recognition mode enabled. The raw data from mass spectrometry sequencing has been uploaded to the iProX database (PXD044165) (https://www.iprox.cn//page/project.html?id=IPX0006791000).

## Immunoprecipitation

For immunoprecipitation, HeLa cells were infected with NDV or transfected with the pCAGGS-eIF4A1-HA, pCAGGS-HNP-his, pCAGGS-INP-his or pCAGGS-his plasmid. To perform co-immunoprecipitation, HeLa cells were co-transfected with HA tagged eIF4A1, and his-tagged HNP or INP plasmids. Cells were harvested, washed and lysed in NP-40 lysis buffer (Beyotime) containing 1 mM phenylmethylsulfonyl fluoride (PMSF) and 1 mg/mL protease inhibitor cocktail (Roche), then centrifuged at 12,000 ×g for 15 min. Total protein was

incubated overnight with specific primary antibodies and the complexes were pulled down by incubation with protein-A+G agarose beads for 2 h. The beads were washed with NP-40 lysis buffer for 3 times, resuspended in 2×SDS protein loading buffer, boiled, and subjected to SDS-PAGE. Immunoprecipitated proteins were detected by Western blot analysis. Run-off lysates were used for Western blot to detect β-actin for sample normalization.

### GST pull-down assay

The NDV NP gene, subcloned into the pET-N-GST vector, was introduced into Escherichia coli BL21 (DE3) for expression. The GST-NP fusion protein was subsequently isolated using glutathione-Sepharose beads (Beyotime), following the manufacturer's protocol. To provide a brief overview, GST or GST-NP cells were coupled with glutathione beads and exposed to HeLa cell lysates containing HA-eIF4A1 for a 2-hour incubation at 4˚C. After washing to eliminate unbound molecules, the protein complexes were released using loading buffer and identified through a Western blot analysis employing anti-GST and anti-HA antibodies.

### Construction of HeLa cell lines for stable expression of NP protein

Referring to a previous method [39], plasmids including pWPXL (12257), pMD2.G (12259), and psPAX2 (12260) were obtained from Addgene (Cambridge, MA, USA). The NP cDNA of Herts/33 and $I_4$ were cloned into the BamHI and EcoRI sites of the pWPXL vector. The lentivirus was prepared in 293T cells and titrated as previously described [40]. NDV NP protein expression was detected by Western blotting. Primer sequences are provided in S8 Table.

### RNA-Seq and RNC-Seq

We followed the RNA-Seq and RNC-Seq (Ribosome Nascent-chain Complex sequencing) protocols as previously described [38]. Briefly, control HeLa cells and HeLa cells stably expressing HNP and INP were seeded onto 10 cm dishes and allowed to grow for 18 h. Half of the cells from each dish were collected for total RNA extraction, while the other half were subjected to ribosomal isolation. RNA was extracted from the ribosomal fraction, which combined the 80s fraction and the polyribosomal fraction. Library construction and sequencing were performed by the Beijing Genome Institute (www.genomics.org.cn, UW Genome, Shenzhen, China). The raw sequence data were deposited in the NCBI Sequence Read Archive (https://www.ncbi.nlm.nih.gov/geo/) under accession number GSE233617. After obtaining the reads, we normalized them to RPKM using the RESM software to analyze gene expression. Differential gene expression was determined based on a log2-ratio absolute value $\geq 1$ and FDR $\leq 0.001$, as defined by the UW Genome Bioinformatics Service. The high-throughput data has been uploaded to the GEO database (GSE233617) (https://www.ncbi.nlm.nih.gov/geo/query/acc.cgi?acc=GSE233617).

### Enzyme-linked immunosorbent assay for IFN-α and IFN-β

HeLa cells were seeded into 12-well plates and infected with $rI_4$ or rHerts/33 at 1MOI and 10MOI, respectively. At 24hpi, 0.4mL of culture supernatant was collected, and the levels of IFN-α or IFN-β were measured using the V Human interferonα (IFN-α) ELISA kit (HM10251) and Human interferonβ (IFN-β) ELISA kit (HM10099) following the manufacturer's instructions. The protein expression levels of IFN-α or IFN-β (pg/mL) were calculated based on the standard curve generated during the assay.

### LC-MS/MS analysis

HeLa cells were transfected with pCAGGS-HNP-His or pCAGGS-INP-His. After 48 hours, the cells were lysed using NP-40, and 5 μg of rabbit IgG along with protein A+G beads were co-incubated with the cell lysates for 4 hours to remove non-specific binding. Subsequently, co-IP experiments were conducted using rabbit anti-NP polyclonal antibody, and the precipitated complexes were subjected to mass spectrometry analysis by Applied Protein Technology (APTBIO, China, Shanghai) 10 cm long, 75 μm inner diameter, 3μm resin) in buffer A (0.1% Formic acid) and separated with a linear gradient of buffer B (84% acetonitrile and 0.1% Formic acid) at a flow rate of 300 nl/min controlled by IntelliFlow technology. LC-MS/MS analysis was performed on a Q Exactive mass spectrometer (Thermo Scientific) that was coupled to Easy nLC (Proxeon Biosystems, now Thermo Fisher Scientific) for 60/120/240 min (determined by project proposal). The mass spectrometer was operated in positive ion mode. MS data was acquired using a data-dependent top10 method dynamically choosing the most abundant precursor ions from the survey scan (300–1800 m/z) for HCD fragmentation. Automatic gain control (AGC) target was set to 3e6, and maximum inject time to 10 ms. Dynamic exclusion duration was 40.0 s. Survey scans were acquired at a resolution of 70,000 at m/z 200 and resolution for HCD spectra was set to 17,500 at m/z 200, and isolation width was 2 m/z. Normalized collision energy was 30 eV and the underfill ratio, which specifies the minimum percentage of the target value likely to be reached at maximum fill time, was defined as 0.1%. The instrument was run with peptide recognition mode enabled. The raw data from mass spectrometry sequencing has been uploaded to the iProX database (PXD044165) (https://www.iprox.cn//page/project.html?id=IPX0006791000).

### Statistical analysis

The band intensities of Western blot were calculated using ImageJ software (NIH). Statistical significance was determined by one-way analysis of variance (ANOVA) test or two-way analysis of variance (ANOVA) by using GraphPad Prism 8.0 software (GraphPad Software). Data were considered statistically significant at $p < 0.05$. Values were expressed as mean ± standard error of the mean.

## Results

### The infectivity of several genotype VII NDVs on tumor cell lines is extremely poor

Our previous studies revealed that genotype VII NDVs have greater infectivity on avian immune cells in vitro experiments [25–27]. In this study, we investigated the replication ability of multiple NDV strains of different genotypes, including La Sota (genotype II), Mukteswar and JS7122 (genotype III), Herts/33 (genotype IV), $I_4$, JS1816, YN1200, and ZJ1 (genotype VII), and F48E8 (genotype IX), by measuring the 50% tissue culture infective dose ($TCID_{50}$) on different tumor cell lines.

According to the results, all genotype VII NDVs showed significantly weaker replication, with a $TCID_{50}$ value of 1–2 lg$TCID_{50}$/0.1mL, about $10^6$-fold lower than other genotypes on five tumor cell lines (Fig 1A–1E). To confirm this replication defect, we selected ten additional NDV strains of genotype VII to test their replication ability in HeLa cells. The result showed they also exhibited low replication capacity (Fig 1F). Interestingly, we observed the same biological phenomenon in various non-tumor mammalian cell lines (S1 Fig). However, the difference in infectivity among NDV strains is more pronounced in tumor cells compared to non-tumor mammalian cells.

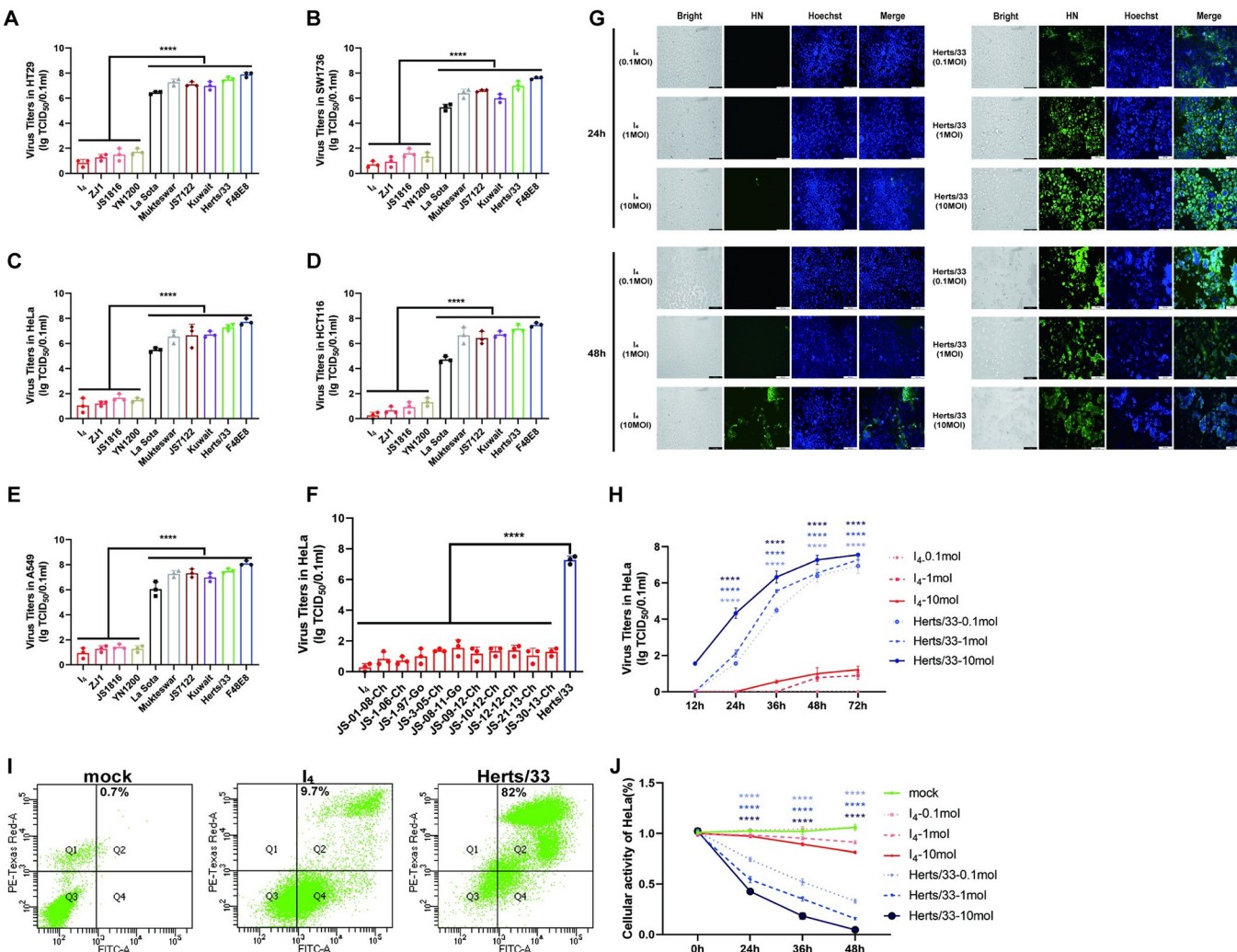

**Fig 1. Infectivity of different genotypic strains on tumor cells.** Viral titers at 72hpi in (A) HT29, (B) SW1736, (C) HeLa, (D) HCT116, (E) A549. (F) TCID$_{50}$ value of ten additional genotype VII NDV strains in HeLa cells. Representative data, shown as the mean ± SD (n = 3), were analyzed with one-way ANOVA. ****, $P<0.0001$, (G) IFA experiments were performed with anti-HN and Hoechst at 24h and 48h after infection of NDVs with 0.1MOI, 1MOI, and 10MOI. (H) Replication of I$_4$ and Herts/33 in HeLa cells. Cells were infected with I$_4$ and Herts/33 at 0.1MOI, 1MOI, and 10MOI for 1 h at 37°C. Viral titers in the supernatant were expressed using TCID$_{50}$ value. (I) Apoptosis was detected by flow cytometry at 24h after infection at 10MOI, yielding apoptotic cells as a percentage of the total cell count. (J) The oncolytic effect of NDVs on HeLa cell line in vitro. HeLa cells were infected with 0.1MOI, 1MOI, and 10MOI at 0, 24, 36, and 48h. The cell viability was measured by CCK-8 assay and expressed as a percentage relative to the control group, and results are shown as the mean of three independent experiments. Representative data, shown as the means ± SDs (n = 3), were analyzed with two-way ANOVA. (****, $P < 0.0001$).

Subsequently, genotype VII NDV strain I$_4$ and genotype IV strain Herts/33 were selected to compare their oncolytic ability. According to the indirect immunofluorescence assay and the viral growth curve, I$_4$ was barely detectable even at 10MOI in HeLa cells, while Herts/33 could replicate at 0.1MOI effectively (Fig 1G and 1H). Additionally, we found that I$_4$ infection slightly affected cellular viability and apoptosis even at 10MOI in HeLa cells (Fig 1I and 1J). In contrast, infection of Herts/33 at 0.1MOI reduced HeLa cellular viability to 39%. Moreover, Herts/33 infection induced apoptosis in approximately 82% of HeLa cells (Fig 1I). Collectively, our findings demonstrate that several genotype VII NDVs exhibit poor infectivity in tumor cells.

## NP protein is responsible for oncolytic deficiency of several genotype VII NDVs

To identify the critical gene(s) responsible for the oncolytic activity of NDV, we constructed a series of replacements between genotype VII NDV strain $I_4$ and genotype IV NDV strain Herts/33 by exchanging the genes encoding NP, P, M, F, HN, and L. These proteins' amino acid sequence homology was NP: 91.64%, P: 87.29%, M: 88.21%, F: 88.81%, HN: 86.83%, and L: 89.28%. Recombinant viruses were all successfully rescued.

We first assessed the replication ability of recombinant viruses with simultaneous replacement of NP, P, and L genes on four tumor cell lines. As shown in Fig 2A, their strong virulence properties measured by ICPI indicated that the biological functions of the structural proteins of Herts/33 and $I_4$ are compatible. The viral titer of rI-NPPL$^H$ was significantly increased to approximately 8 $\log_{10}$TCID$_{50}$/0.1mL, while the titer of rH-NPPL$^I$ was significantly decreased to approximately 2 $\log_{10}$TCID$_{50}$/0.1mL, compared to their corresponding parental strains (Fig 2B and 2C). At 24 hpi, trace amounts of NP and HN proteins from rI$_4$ and rH-NPPL$^I$ were also detected (Fig 2D). These results indicated that the NP, P, and L genes might play critical roles in the oncolytic defection of genotype VII NDVs.

We subsequently investigated the impact of individual gene replacement of NP, P, and L genes on the oncolytic capacity. We found that only the TCID$_{50}$ values of rI-NP$^H$ and rH-NP$^I$ were significantly increased or decreased compared to their corresponding parental strains, whereas those of rI-P$^H$, rI-L$^H$, rH-P$^I$, and rH-L$^I$ showed no significant changes (Fig 2E and 2F). At 24hpi, trace amounts of NP and HN proteins from rI$_4$ and rH-NP$^I$ were also detected, but high levels of these were detected from rHerts/33 and rI-NP$^H$ (Fig 2G and 2H). All recombinant viruses also showed high viral titers on CEF, further indicating the biological

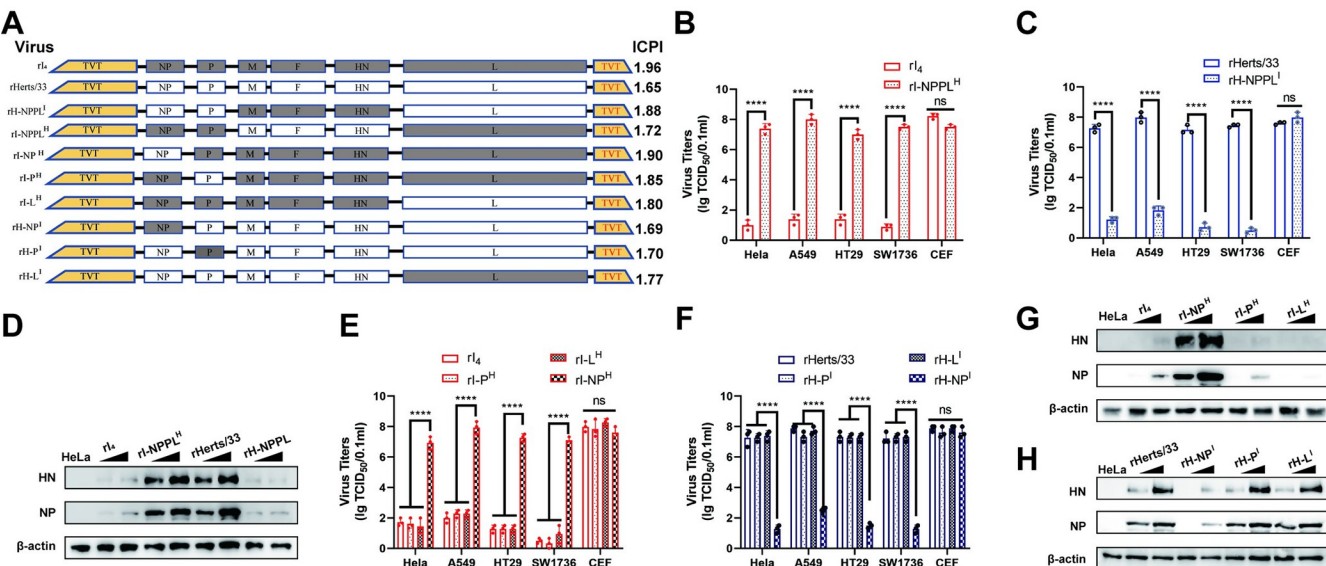

**Fig 2. NP is the main protein responsible for the phenotype.** (A) Schematic diagram of the cloning strategy for replacement of NP, P, and L genes between rHerts/33 and rI$_4$. The construction strategy is described in the S1 Table. The virulence of the different recombinant viruses was determined by measuring the ICPI in 1-day-old chickens. (B and C) TCID$_{50}$ value of the virulent strains after simultaneous replacement of NP, P, and L at 72hpi on tumor cell lines. (D) Expression of viral proteins on HeLa cells by recombinant viruses after simultaneous replacement of NP, P, and L. Western blot analysis was performed by anti-NP and anti-HN at 24h after infection with NDVs at 1MOI and 10MOI, respectively. (E and F) TCID$_{50}$ value of the virulent strains after individual gene replacement of NP, P, or L at 72hpi on tumor cell lines. (G-H) Viral proteins expression on HeLa cells by recombinant viruses after individual gene replacement of NP, P, or L. Western blot analysis was performed by anti-NP and anti-HN at 24h after infection with NDVs at 1MOI and 10MOI, respectively. Representative data, shown as the means ± SDs (n = 3), were analyzed with two-way ANOVA. ****, $P<0.0001$.

compatibility of the structural proteins. These results suggest that the NP protein is the critical determinant of NDV's oncolytic activity. The absence of significant differences in the viral titers among rI-NPP[H], rI-NPL[H], and rI-NP[H] implies that there is no synergistic effect among the homologous NP, P, and L proteins, and this was confirmed with reverse verification (S2B and S2C Fig). In summary, our results demonstrate that NP protein is the decisive factor for NDV infection of tumor cells.

## Viral infectivity on tumor cells is notably affected by the 450th amino acid of the NP protein

The NP protein of paramyxoviruses consists of two domains: the conserved N-core domain and the variable N-tail domain [5,6,8,10]. Disordered structural domains are widely reported in their N-tail domain, which lacks stable secondary and tertiary structures, making it challenging to conduct homologous modeling for comparison. Therefore, we predicted the disordered structural domains of NP protein from rI$_4$ and rHerts/33 as previously described (S3A–S3D Fig) [41]. The N-tail domain's first intrinsically disordered region (IDR) is located at approximately amino acids 380 to 440, and the second IDR is located at approximately 450 to 489 (Fig 3A). Recombinant viruses were constructed based on these predictions (Fig 3B).

To pinpoint the critical domain responsible for the oncolytic activity of NDV, we generated a series of recombinant viruses by replacing the N-core and N-tail of the NP protein between rI$_4$ and rHerts/33. As depicted in Fig 3C, rI-Ntail[H], which contains the N-tail domain of rHerts/33, displayed a $10^5$-fold increase in TCID$_{50}$ value on tumor cell lines compared to the corresponding parental rI$_4$ strains. rH-Ntail[I], which contains the N-tail domain of rI$_4$, exhibited a $10^5$-fold decrease in TCID$_{50}$ value on tumor cell lines compared to rHerts/33 (Fig 3D). In contrast, the infectivity of recombinant viruses constructed by exchanging the N-core of rI$_4$ or rHerts/33 showed no significant changes relative to their parental strains (S3F and S3G Fig), indicating that the N-tail is critical for the oncolytic activity of NDV. Subsequent infectivity experiments demonstrated that rH-NP$_{450-489}$[I] and rI-NP$_{450-489}$[H], which were constructed by exchanging the second IDR of the N-tail domain, exhibited a $10^5$-fold reduction or elevation in TCID$_{50}$ value on tumor cell lines relative to their parental strains (Fig 3E and 3F). However, the infectivity of recombinant viruses constructed by exchanging the first IDR of the N-tail domain did not exhibit significant changes (S3H and S3I Fig). Additionally, rI$_4$, rH-Ntail[I], and rH-NP$_{450-489}$[I] exhibited almost no NP and HN proteins at 24hpi, but high levels were detected in Herts/33, rI-Ntail[H], and rI-NP$_{450-489}$[H] (Fig 3G). These results suggest that the second IDR of N-Tail is essential for the oncolytic activity of NDV.

Our analysis of the second IDR identified three conserved differential sites in genotype VII NDVs: positions 450, 464, and 479 (Fig 3I). The F450L mutation of NP protein significantly increased the infectivity of rI$_4$, while the L450F mutation significantly decreased the infectivity of rHerts/33. The other two mutations did not affect NDV's infectivity in HeLa cells. However, it's worth noting that the TCID$_{50}$ value of rI-NP$_{F450L}$ is still nearly 10 times lower compared to that of rHerts/33. These results suggest that the amino acid residue at position 450 is critical for NDV's oncolytic activity. To determine whether this effect is valid for other genotypes, we rescued rK-NP$_{L450F}$ and rL-NP$_{R450F}$. Infectivity tests showed that rI-NP$_{F450L}$ exhibited a $10^{4-6}$-fold enhancement in TCID$_{50}$ value compared to the rI$_4$ strain (Fig 3J), while rH-NP$_{L450F}$, rK-NP$_{L450F}$, and rL-NP$_{R450F}$ showed approximately a $10^{4-6}$-fold reduction in TCID$_{50}$ value relative to rHerts/33, rKuwait, and rLa Sota (Fig 3K–3M) on four tumor cell lines. Furthermore, rI$_4$, rH-NP$_{L450F}$, and rK-NP$_{L450F}$ were difficult to detect, while rI-NP$_{F450L}$, rHerts/33, and rKuwait could be detected at 12 hpi at either 1MOI or 10MOI (Fig 3N). Interestingly, amino acid 450 of the NP protein also appeared to have a significant effect on NDV in a variety of

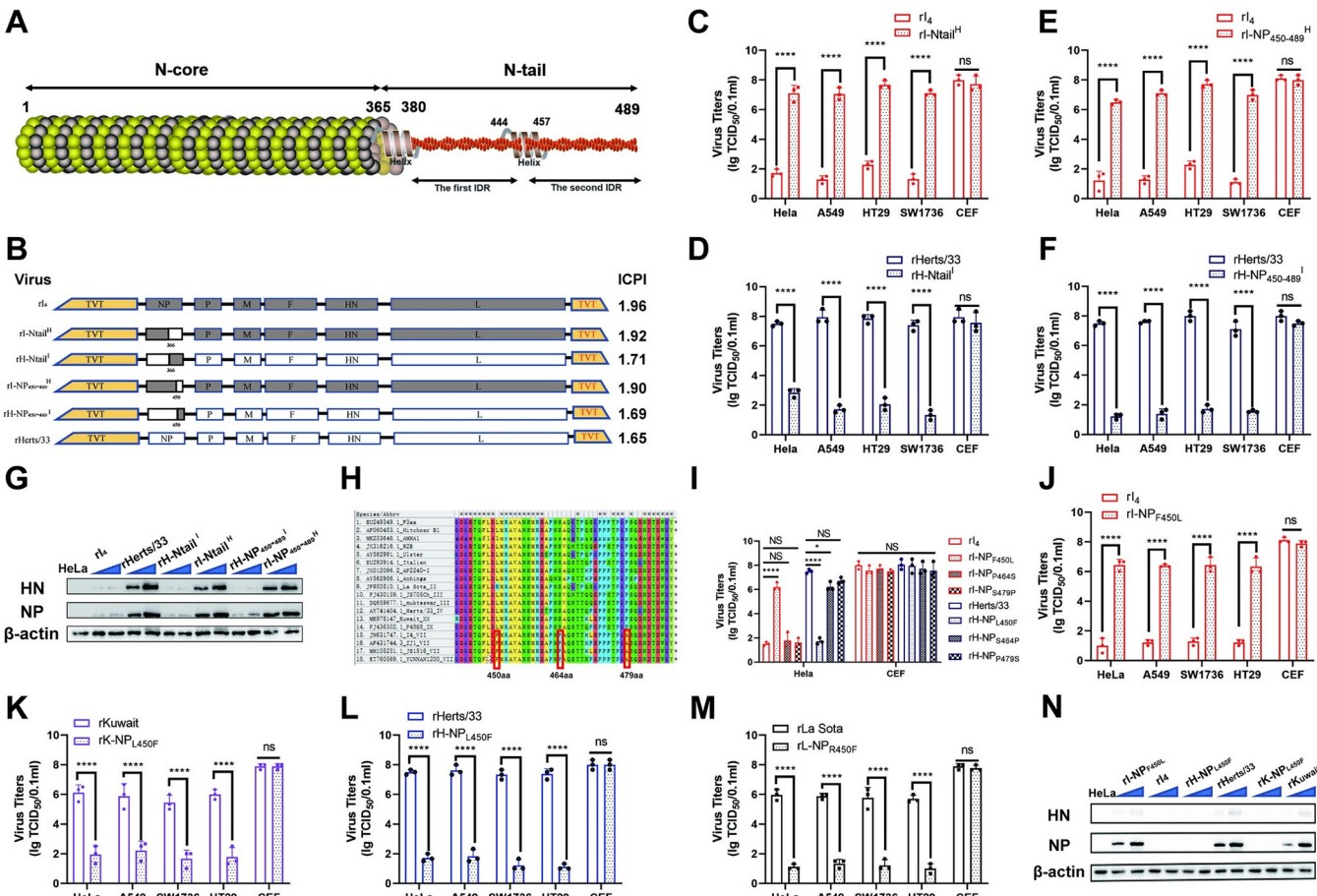

**Fig 3. The 450th amino acid of the NP protein determines the infectivity of NDVs on tumor cells.** (A) Structural diagram of the NP protein. (B) Schematic diagram of the cloning strategy for exchanging the whole N-tail and the second IDR of the N-tail domain between rHerts/33 and rI$_4$. The virulence of the different recombinant viruses was determined by measuring the ICPI in day-old chickens. (C, D, E, and F) TCID$_{50}$ value of recombinant strains after the whole N-tail and the second IDR of the N-tail domain is replaced at 72hpi on tumor cell lines. (G) Expression of viral proteins on HeLa cells by recombinant viruses. Western blot analysis was performed by anti-NP and anti-HN at 24h after infection with NDVs at 1MOI and 10MOI, respectively. (H) Schematic representation of the comparison of NP genes from different virulent strains by MEGA. (I) TCID$_{50}$ values of NDVs at 72hpi in HeLa cells and CEF cells after mutation of each conserved site in rHerts/33 and rI$_4$. (J, K, L, M) TCID$_{50}$ values of mutant NDVs at 72hpi on tumor cell lines. (N) Expression of viral proteins on HeLa cells by virulent mutant strains. Western blot analysis was performed by anti-NP, anti-HN, and anti-β-actin at 12h after infection with NDVs at 1MOI and 10MOI, respectively. All experiments were repeated thrice, and results are expressed as mean ± SDs. Two-way ANOVA was used to evaluate the significance of differences. ****, $P < 0.0001$.

mammalian cells (S3F–S3I Fig). In conclusion, our findings indicate that the 450th amino acid residue of NP plays a crucial role in the infection of NDV on tumor cell lines.

## NDVs with Phe at the 450th amino acid position of NP protein are mainly genotype VII strains

To explore the distribution of NDVs containing phenylalanine (F) at position 450 of the NP protein and screen the genotypes suitable for oncolytic vaccine, we downloaded 890 full-length NDV sequences from NCBI and constructed a phylogenetic tree based on the complete sequence of the F gene.

As shown in Fig 4A, phylogenetic analysis showed the distribution of NDV subtypes, with 161 Class I and 729 Class II strains. The class II NDVs were grouped into 18 genotypes, i.e., genotype I (68/890), genotype II (66/890), genotype III (8/890), genotype IV (7/890), genotype

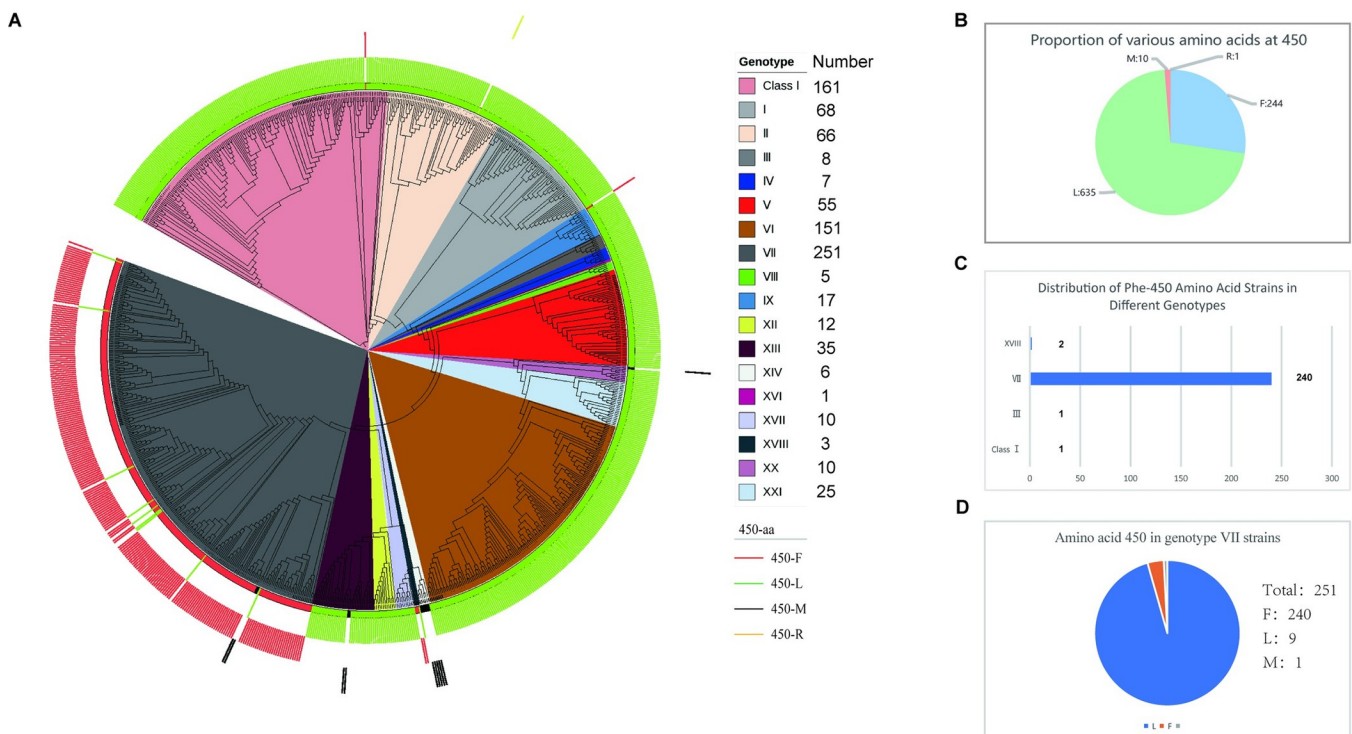

**Fig 4. Viruses containing phenylalanine residues at 450 of NP are primarily found in genotype VII.** (A) ML phylogenetic tree of 890 NDV strains based on full-length F-gene sequences. The evolutionary tree was constructed using PhyloSuite in a SYM model. Pie chart representing the number of various residues at position 450 of amino acids for 890 NDV strains. (B) The pie chart illustrates all possible amino acids at position 450 of the NP protein and their proportions. (C) The pie chart shows the distribution of 450aa-phe-NP strains in each genotype. (D) The pie chart displays all the possibilities and their frequencies of the 450th amino acid position of the genotype VII strains' NP protein.

V (55/890), genotype VI (151/890), genotype VII (251/890), genotype VIII (5/890), genotype IX (17/890), genotype XII (12/890), genotype XIII (35/890), genotype XIV (6/890), genotype XVI (1/890), genotype XVII (10/890), genotype XVIII (3/890), genotype XX (10/890) and genotype XXI (25/890). Of all the strains, 244 were F at position 450, 635 were L, 10 were M, and 1 was R (Fig 4B). Among the 450-F-NP strains, there were 240 strains of genotype VII NDV, 1 strain of Class I NDV, 2 strains of genotype XVIII NDV and 1 strain of genotype III NDV (Fig 4C). Notably, over 95% of genotype VII NDVs (240/251) exhibited the 450-F-NP phenotype. (Fig 4D).

## The replication discrepancy arises during the process of viral mRNA translation

To investigate the process of the observed differences, we examined the ability of the NDVs to adsorb and bind to cells by flow cytometry (Fig 5A). The results indicate no significant difference in virus adsorption between the $rI_4$ and rHerts/33 strains. Additionally, similar nucleocapsid levels entered into cells were shown by the similar copies of gRNA between $rI_4$ and rHerts/33 at 0 hpi (Fig 5B). However, at 4 hpi, the gRNA, cRNA, and mRNA expression of the rHerts/33 was significantly higher than that of $rI_4$ (Fig 5B–5D), suggesting that the observed differences may occur during the viral replication process.

To investigate this possibility, relative minigenome were constructed and used to compare the $rI_4$ and rHerts/33 polymerase activity. The mRNA copies and genomic RNA copies of GFP from different minigenome systems were found to be at similar levels, indicating no significant

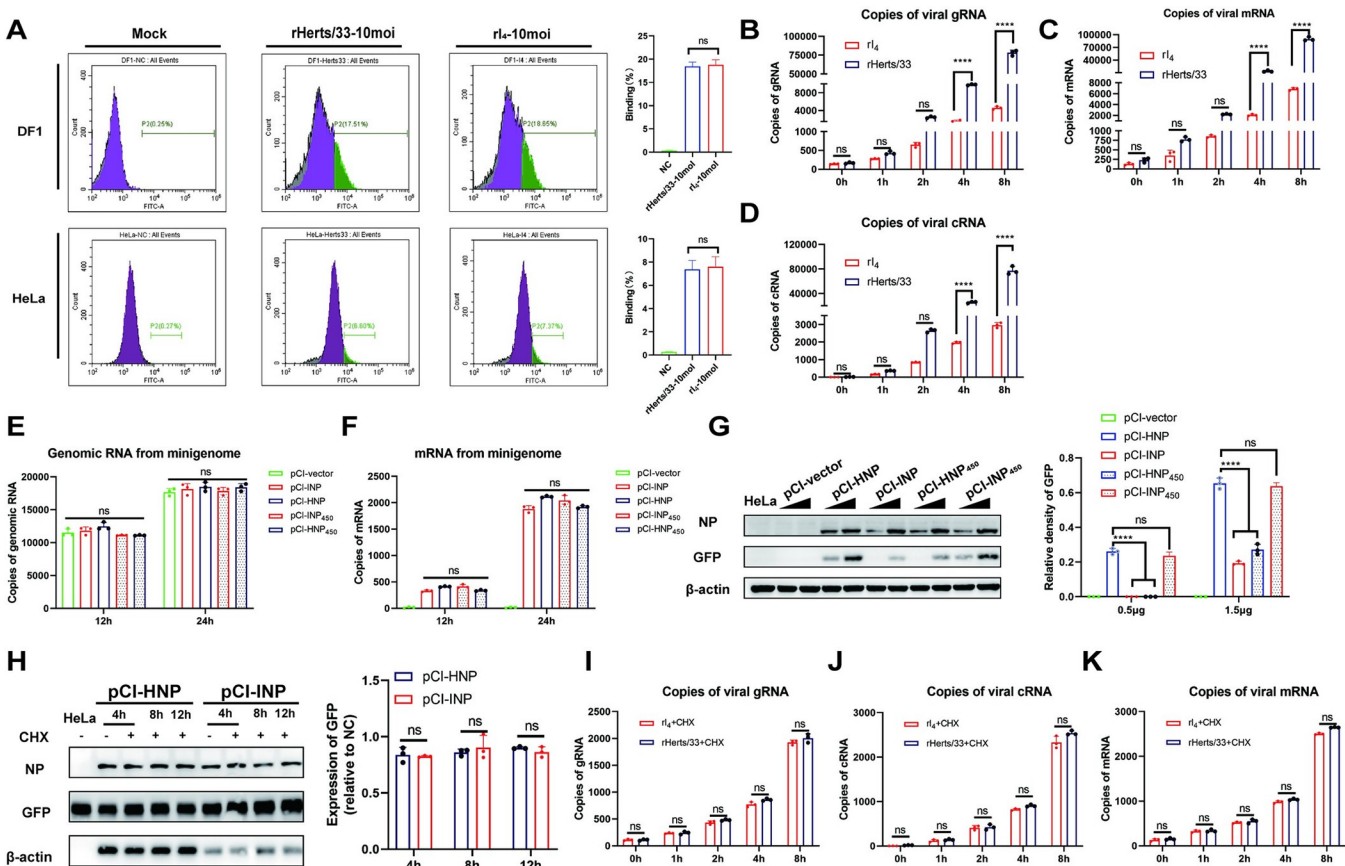

**Fig 5. Variations in viral replicative capability occur during the translation of viral mRNA.** (A) HeLa cells and DF1 cells were infected with NDVs at 10MOI at 37˚C for 1h and then incubated with anti-HN mouse monoclonal antibody and goat anti-mouse IgG/FITC at 4˚C. After that, cells were washed and assessed by flow cytometry. (B, C, D) HeLa cells were infected with NDV (10MOI) at 37˚C for 0.5h and then were collected at 0h, 1h, 2h, 4h, and 8h. Total RNA was extracted and reverse-transcribed using specific primers for gRNA (B), mRNA (C), and cRNA (D) of NDVs. Copy numbers were determined using quantitative RT-PCR. (E and F) Total cellular RNA was extracted at 12h and 24h after transfection of 1.5 μg minigenome into HeLa cells. Reverse transcription was performed using specific primers to detect genomic RNA (E) and mRNA (F) of GFP by quantitative RT-PCR. (G) Expression of GFP was detected at 24h in HeLa cells after transfecting 0.5μg or 1.5μg minigenome with anti-GFP, anti-NP, and anti-β-actin. (H) After transfection with different minigenome systems for 24h, cells were treated with 100μg/ml CHX, and then cells were harvested at 4, 8, and 12 hours. Expression of GFP and NP was detected with anti-GFP, anti-NP, and anti-β-actin. (I, J, K) HeLa cells were treated with 100μg/ml CHX for 30mins and then infected with NDV (10MOI) at 37˚C for 0.5h. After that, cells were collected at 0h, 1h, 2h, 4h, and 8h. Total RNA was extracted and reverse-transcribed using specific primers for gRNA (I), mRNA (J), and cRNA (K) of NDVs. Copy numbers were determined using quantitative RT-PCR. Data are presented as means from three independent experiments. Significance is analyzed by two-way ANOVA (****, $p < 0.0001$).

difference in polymerase activity between different NDV strains (Fig 5E and 5F). However, the expression of GFP protein in HeLa cells was higher in the HNP and INP$_{450}$ groups than in the INP and HNP$_{450}$ groups (Fig 5G), while those were similar in CEF cells detected by Western Blot (S4 Fig). Treatment with CHX did not result in significant degradation of NP and GFP in cells (Fig 5H), which ruled out the possibility of protein degradation and suggested that the observed differences in protein levels may occur during viral mRNA translation. In addition, treatment with CHX eliminated the difference in the levels of viral gRNA, cRNA, and mRNA during the first 8 hours after infection with rI$_4$ or rHerts/33 (Fig 5I–5K), supporting our hypothesis that the observed differences in protein expression are likely due to differences in mRNA translation. Our results suggest that NP's 450th amino acid residue may play an essential role in the translation of viral mRNA.

## The 450th amino acid of NP influences the loading of GFP mRNA in the minigenome systems onto ribosomes

To investigate the potential involvement of NP in the regulation of viral mRNA, HeLa cells were transfected with different minigenome systems. Each system contained TVT-HGFP, pCI-HP, and pCI-HL with different sources of NP (pCI-HNP or pCI-INP) as the only variable, while empty vector was used to maintain consistent transfection plasmid amounts. Cell extracts from transfected cells were subjected to 10–50% sucrose density gradient ultracentrifugation, and the fractions were analyzed by absorbance at 254nm. The highest absorbance at fraction 6 and the peak of rpS6 at fractions 5–8 indicated that fractions 5–12 contained ribosomes. Protein in half of each fraction's volume was subjected to TCA precipitation and analyzed by Western blot to assess the distribution of NP protein. INP, HNP, $INP_{450}$, and $HNP_{450}$ were all detected in the polyribosomal fractions. However, HNP and $INP_{450}$ localized to the ribosomes at significantly higher levels than INP and $HNP_{450}$ (Fig 6A). RNA was extracted from the other half of each fraction, and quantitative analysis of GFP mRNA was performed. The results showed that the copies of GFP mRNA in each polyribosomal fraction were significantly lower in the INP and $HNP_{450}$ groups (Fig 6B), which was only about 10% of that in the HNP and $INP_{450}$ groups. Additionally, the GFP mRNA levels were similar among the groups in the total RNA (Fig 6C). These data demonstrate that $450^{th}$-L-NP enhances the translation of GFP mRNA transcribed by viral polymerase.

After discovering the difference in the ability of HNP and INP to recruit ribosomes, we constructed a series of NP truncated plasmids to explore the functional domains (Fig 6D). Cells transfected with various plasmids were harvested. To eliminate the impact of varying RNA levels on polyribosome enrichment, RNase was added to the cell lysate. Subsequently, the cell lysate underwent ribosome isolation, following the method described above. $HNP_{1-245}$ and $HNP_{245-366}$ were not detected in the ribosomal fraction, while $HNP_{245-489}$, $HNP_{366-489}$, and $HNP_{122-366}$ could be detected, indicating that amino acids at positions 366–489 and 122–366 contain intact structural domains localized to the ribosome (Fig 6E). These results suggested that NP has multiple structural domains involved in the co-localization of ribosomes and that amino acid position 450 is one of the critical sites.

## The 450th amino acid of the NP protein impacts the loading of viral mRNA onto ribosomes

Given the observed correlation between the 450th amino acid of the NP protein and the translation of GFP mRNA transcribed by the polymerase in the minigenome system, our next objective was to explore the association between the 450th amino acid of NP protein and the translation of viral mRNA during viral infection. We found a significant decrease in absorbance at A254nm after infection with $rI_4$ and rHerts/33 at 10MOI, indicating severe translation defects in both strains at 12hpi (Fig 7A). Quantitative RT-PCR analysis revealed that only about 13% of the viral mRNA in $rI_4$ and $rH-NP_{L450F}$-infected cells were bound to ribosomes, while it reached 67% in rHerts/33 and $rI-NP_{F450L}$-infected cells (Fig 7B–7C). The mRNA copies of IFNα, IFNβ and β-actin were significantly higher in the polysomal fraction of the $rI_4$ and $rH-NP_{450}$ group while significantly lower in that of the rHerts/33 and $rI-NP_{F450L}$ group (Fig 7D–7F). This leads to more pronounced phosphorylation of STAT1 and STAT2, as well as significant upregulation of RIG-I, PKR, STAT1, and STAT2 within cells infected with $rI_4$ (Fig 7G). The heightened activation of p-IRF3 by Herts/33 may be attributed to the substantial production of viral dsRNA (Fig 7G). Consequently, the mRNA expression levels of IFNα and IFNβ were markedly higher in the rHerts/33 group compared to the $rI_4$ group at 24 hpi (Fig 7H). However, at 24 hpi, the expression of IFNα and IFNβ in $rI_4$-infected cells was significantly

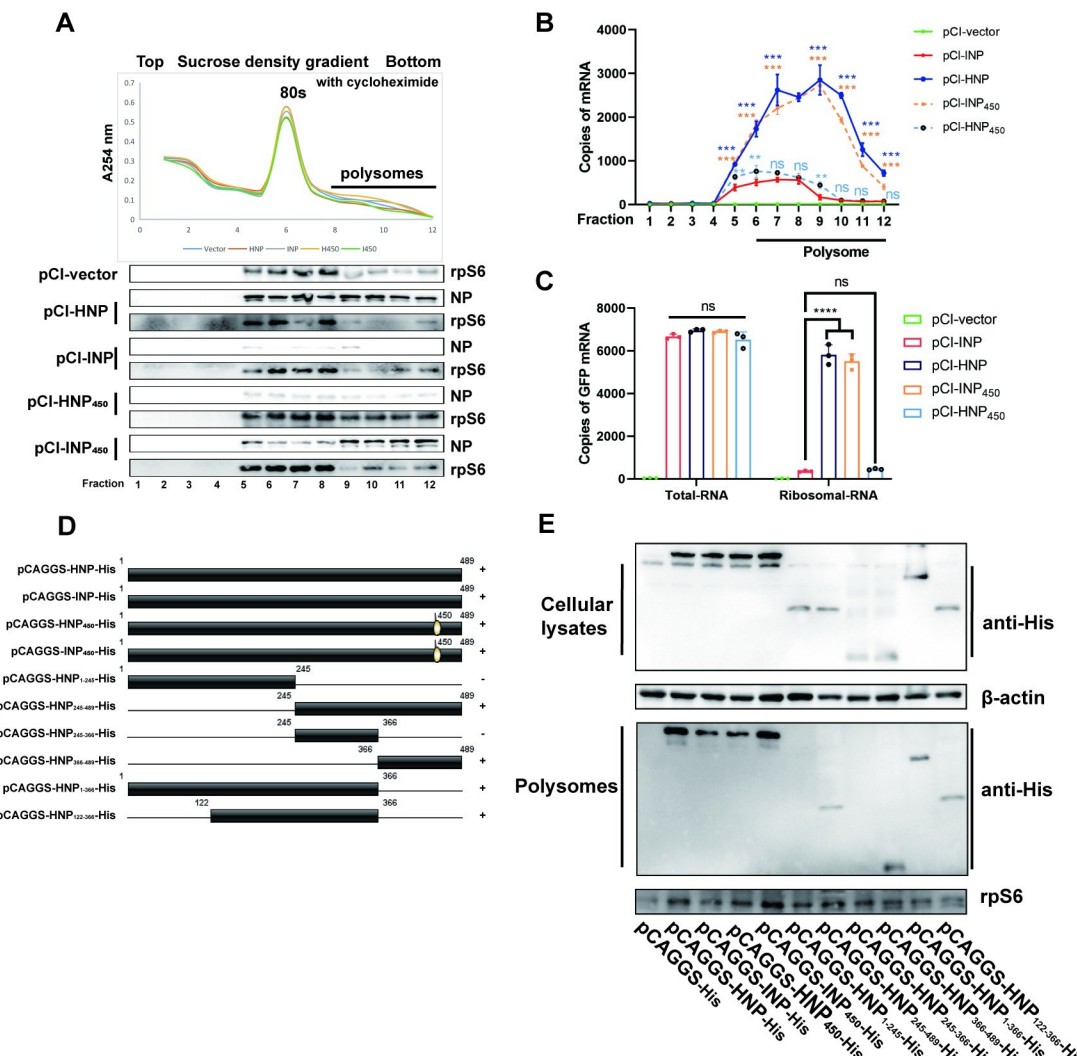

**Fig 6. 450th-L-NP mediates a higher translation efficiency of GFP mRNA from the minigenome.** (A) Cells were collected at 24h after transfection of minigenome or the control plasmid, and half of them were lysed to isolate ribosomes. Top: Total cytoplasmic ribosomes were separated by sucrose density gradient centrifugation, and the absorbance of each fraction was measured at 254nm. Cycloheximide was present in each sample. Lower panel: Protein in half of each fraction's volume was subjected to TCA precipitation and subsequently utilized for immunoblotting with anti-His and anti-rpS6 antibodies. (B) The remaining half of the cells were extracted for total RNA to detect the mRNA of GFP by quantitative RT-PCR. The results represent the mean ± SD of a representative quantitative RT-PCR experiment conducted in triplicate. (C) Samples from the remaining half of each fraction after ribosome isolation were extracted for RNA and assayed for the distribution of mRNA of GFP in complex with ribosomes by quantitative RT-PCR. Results are the mean ± SD of a representative quantitative RT-PCR experiment performed in duplicate three times. Significance was analyzed by two-way ANOVA. (** means $p<0.01$, ***means $p<0.001$). (D) Schematic representation of NP protein deletion mutants. Boxes indicate the protein product of each truncated NP gene, with amino acid positions indicated above the boxes. Straight lines indicate the region of deletion. (E) Residues 122–366 and 366–489 of NP are sufficient for its localization to the ribosome. Multiple c- and n-terminal truncated NPs were expressed in HeLa cells. Cell extracts from transfected cells were subjected to 10–50% sucrose density gradient ultracentrifugation. RNase (100U/mL) was added to the cell lysate to eliminate the impact of varying RNA levels on polyribosome enrichment. Protein in each fraction was subjected to TCA precipitation and subsequently utilized for immunoblotting with anti-His and anti-rpS6 antibodies.

higher than that in the rHerts/33 group (Fig 7I). Knockdown of RIG-I and PKR expression only slightly increased HN expression in the $rI_4$ group and did not restore it to the same level as in the rHerts/33 group (S5 Fig), suggesting that natural immunity and stress responses are

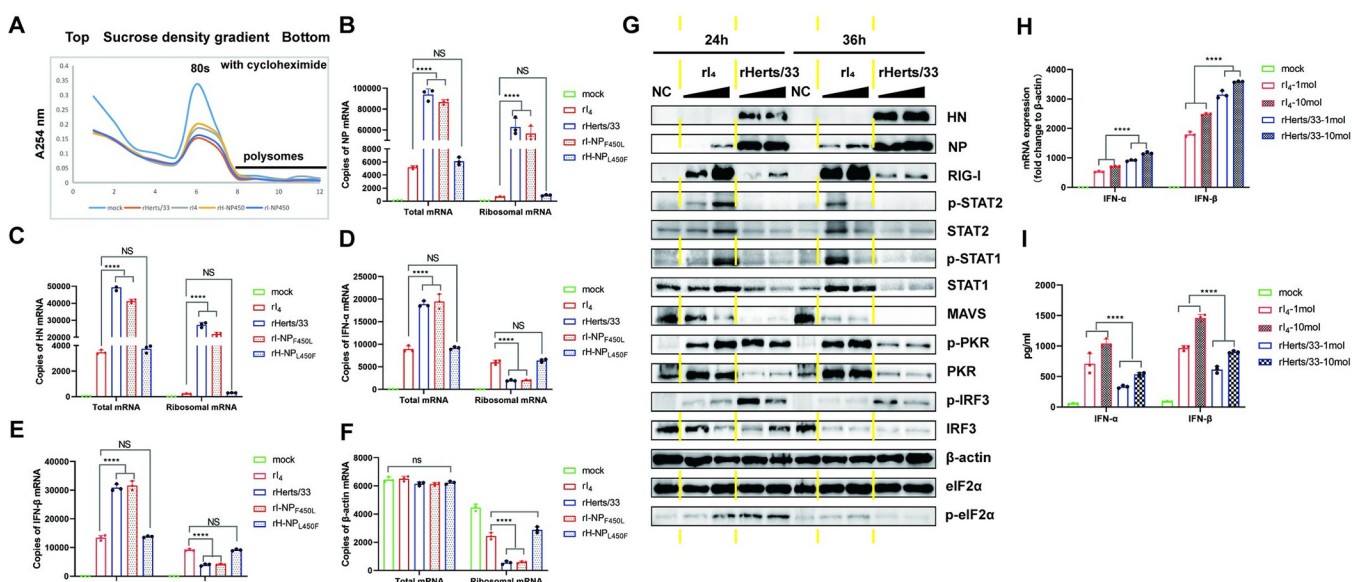

**Fig 7. 450th-L-NP promotes the preferential translation of viral mRNA.** (A) After HeLa cells were infected with NDVs at 10MOI for 12h, ribosomes were isolated, and the absorbance of each component was measured at A254nm. RNA was extracted from each component, and mRNA of NP (B), HN (C), IFN-α (D), IFN-β (E) and β-actin (F) were detected by quantitative RT-PCR. (G) HeLa cells were infected with NDVs at 1MOI or 10MOI, and protein expression was analyzed at 24h and 36h using relative antibodies. (H) qRT-PCR was performed to detect mRNA expression of IFN-α and IFN-β in HeLa cells infected with NDV at 1MOI and 10MOI at 24h. (I) Expression of IFN-α and IFN-β in cell supernatants at 24h was detected by ELISA. Results are shown as the mean ± SD of three independent experiments. And significance was analyzed by two-way ANOVA (****, $p < 0.0001$).

not the decisive factors for NDV replication in tumor cells in this study. Our findings suggest that 450th-L-NP enhances the efficient loading of viral mRNA onto ribosomes during the viral infection process. In contrast, when NDV strains with the 450th-F-NP phenotype replicate, the efficiency of viral mRNA loading onto ribosomes is extremely low, while the efficiency of cellular mRNA loading onto ribosomes can be maintained at a relatively high level.

## HNP interacts with eIF4A1 through 366-489aa

Subsequently, we sought to explore the mechanism by which the NP protein regulates the preferential translation of viral mRNA. We identified proteins that interacted with HNP and INP through mass spectrometry. The raw data has been uploaded to the iProX database with the accession number PXD044165. HNP interacts more extensively with proteins associated with ribosomes and cellular translation (S6 Fig). eIF4A1, one of the components of the translation initiation complex, was found to interact with HNP but not with INP. To determine whether the NP protein interacts with eIF4A1 in NDV-infected cells, NDV-infected cell lysates were subjected to immunoprecipitation using anti-NP antibody and anti-eIF4A1 antibody. Both HNP and eIF4A1 proteins were detected in the precipitated complex (Fig 8A). No INP was detected, indicating an interaction between HNP and eIF4A1 during NDV infection. As INP protein was barely detected during the infection process, we further overexpressed His-tagged HNP and INP in HeLa cells to exclude the impact of protein expression differences on experimental results. Co-IP results demonstrated the formation of a complex between eIF4A1 and HNP but not with INP or the His-tag (Fig 8B). Moreover, we conducted a GST pull-down assay using glutathione beads conjugated to GST-NP or GST-tagged protein to validate NP-eIF4A1 interactions. eIF4A1 from cell lysates was pulled down by GST-HNP but not by GST-INP or GST-bound beads, indicating direct binding between HNP and eIF4A1 (Fig 8C).

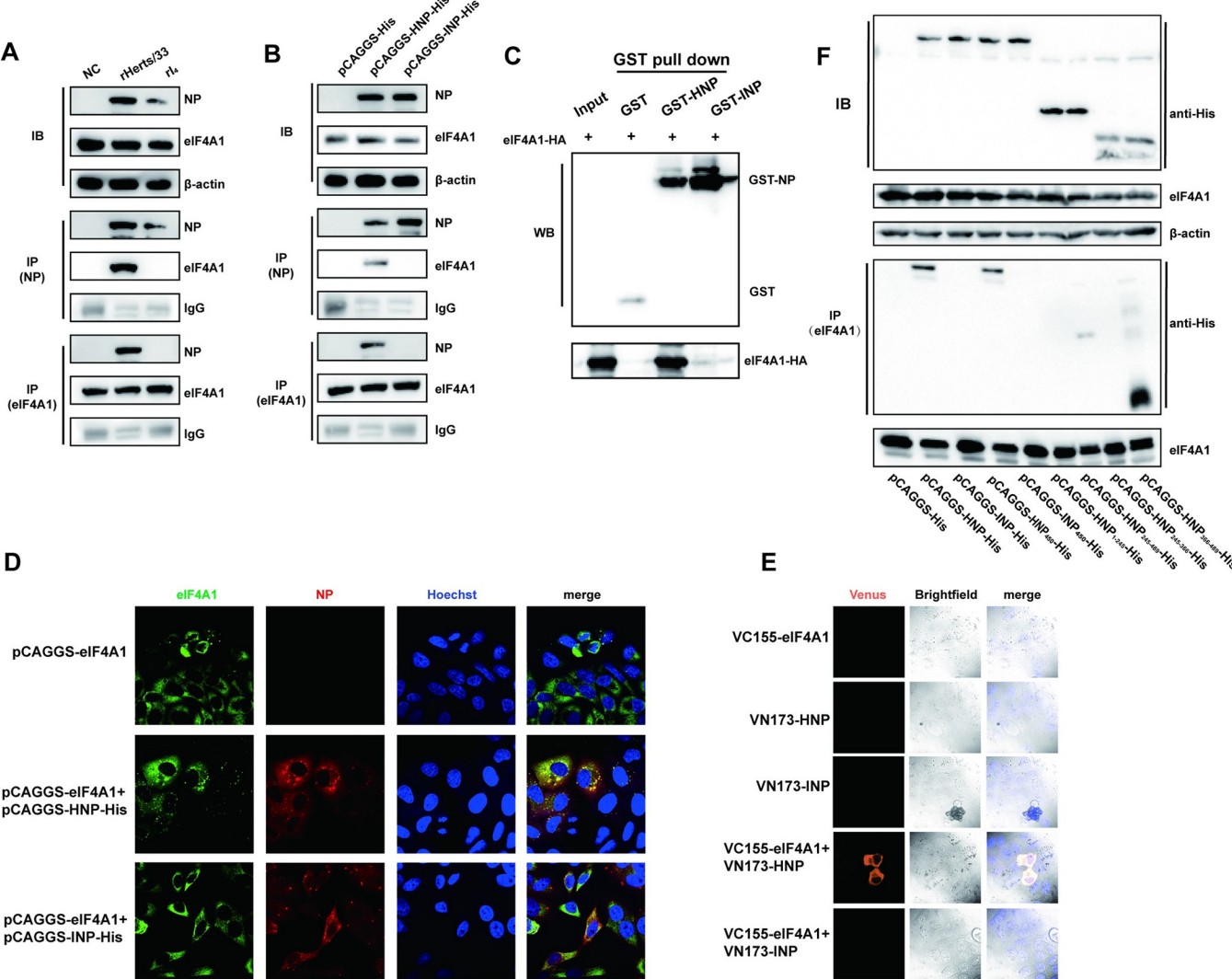

**Fig 8. HNP protein physically interacts with eIF4A1 with 366-489aa.** (A) Co-immunoprecipitation (Co-IP) of NP protein with endogenous eIF4A1 during NDV infection. HeLa cells were infected with rHerts/33, rI₄, or mock-infected and subjected to IP using anti-NP protein or anti-eIF4A1 antibodies. Immunoblotting was performed using the indicated antibodies. (B) HeLa cells were transfected with His-tagged HNP or INP and HA-tagged eIF4A1. After 36 hours, HNP's interaction with eIF4A1 was confirmed through Co-IP using both anti-NP and anti-HA antibodies. (C) GST pull-down assay. Glutathione beads were conjugated with GST or GST-NP fusion protein and incubated with lysate from cells overexpressing eIF4A1. Eluted proteins were subjected to Western Blot, and eIF4A1 presence was detected using anti-HA antibody. GST, GST-HNP, and GST-INP protein expression was confirmed with anti-GST antibody. (D) Redistribution and colocalization of eIF4A1 with HNP protein. HeLa cells were transfected with His-tagged HNP or INP and HA-tagged eIF4A1, fixed at 24 hpi, stained with anti-eIF4A1 and anti-NP antibodies, and visualized using confocal microscopy. (E) Direct interaction between eIF4A1 and HNP confirmed by Bifc assay. Venus luminescence was observed after separate or co-transfection of VC155-eIF4A1, VN173-HNP, and VN173-INP. (F) The C-terminal region 366-489aa of NP is sufficient for heterologous protein association with eIF4A1. Multiple C-terminal and N-terminal truncations of His-tagged NP were expressed in HeLa cells and subjected to IP experiments using anti-eIF4A1 antibody, followed by analysis with specific antibody immunoblotting.

Additionally, we investigated the co-localization of eIF4A1 and NP proteins using confocal fluorescence microscopy. eIF4A1 was observed to have a diffuse distribution throughout the cytoplasm in HeLa cells that overexpressed eIF4A1 alone or co-expressed it with INP (Fig 8D). However, in cells co-expressing HNP and eIF4A1, a significant co-localization between HNP and eIF4A1 was evident. Given previous reports suggesting that NP protein co-localizes with eIF4E, we employed the BiFC assay to validate the interaction between HNP, INP, and eIF4A1. The results showed that only when VN173-HNP and VC155-eIF4A1 were co-

transfected did complete venus fluorescence occur; no other plasmid's single or co-expression produced this effect (Fig 8E). This implies a close and direct interaction between HNP and eIF4A1, whereas there is no evidence of a direct and close interaction between INP and eIF4A1. To determine which NP domain is crucial for its interaction with eIF4A1, we overexpressed several truncated fragments of NDV NP in HeLa cells. Subsequently, the cell was harvested, and Co-IP experiments were conducted. The findings revealed that neither the $HNP_{1-245}$ nor the $HNP_{245-366}$ fragments retained their capacity to bind to eIF4A1 (Fig 8F). Furthermore, it was observed that the 450th amino acid of the HNP protein did not influence its interaction with eIF4A1. Therefore, our findings indicate that amino acids 366–489 of the HNP protein constitute the crucial structural domain for its interaction with eIF4A1.

## HNP promotes NDV replication by inhibiting the eIF4A1/Myc axis

To investigate the impact of eIF4A1 on NDV replication, we employed small interfering RNA (siRNA) to knock down eIF4A1 expression in HeLa cell lines (Fig 9A). The reduction in eIF4A1 expression led to a slight enhancement in replication for both rHerts/33 and $rI_4$ strains (Fig 9A–9D). Following infection of HeLa cells with rHerts/33, we observed a slight activation of p-Akt and a mild downregulation of PDCD4. Surprisingly, the expression of Myc protein, began to decrease from 6 hpi onwards. During this time, NP in the Herts/33 group became detectable, while the expression of eIF4A1 remained unaffected (Fig 9E–9F). This indicates that the infection with rHerts/33 may induce the downregulation of Myc expression. Upon activation of Akt using the Akt activator SC79, we observed an upregulation of Myc, cyclin B1, and cyclin A2 expression, which relies on eIF4A1-mediated translation, in mock and $rI_4$-infected cells (Fig 9G). However, in rHerts/33-infected cells, we detected a downregulation of Myc, cyclin B1, and cyclin A2 mRNA expression (Fig 9H). While overexpressing NP alone did not affect the mRNA expression of Myc, cyclin B1, and cyclin A2 (Fig 9I), to eliminate the variable of mRNA downregulation, we transfected pCAGGS-HNP and pCAGGS-INP into HeLa cells, using pCAGGS as the control, in order to investigate the impact of HNP and INP on the expression of eIF4A1-dependent proteins. Subsequently, in the pCAGGS-HNP group, we noted a decrease in the expression of Myc, cyclin B1, and cyclin A2 (Fig 9J). Hence, our findings imply that the HNP protein could suppressing eIF4A1-associated translation activity. This could be another mechanism by which HNP promotes NDV replication, aside from the functionality of the 450th amino acid.

## HNP inhibits the eIF4A1-dependent translation by positioning itself within the translationally active region

The preferential translation of viral mRNA is often accompanied by the regulation of host mRNA translation. To investigate this phenomenon, we established HeLa cell lines expressing HNP and INP using a lentiviral packaging system. The expression of NP was confirmed through Western Blot analysis (Fig 10A). Subsequently, we conducted RNC-seq. Ribosome-bound RNA was isolated from ribosomal fractions, while total RNA served as a control for assessing RNA abundance changes (S7 Fig). Our findings revealed that both the HNP and INP groups exhibited a similar number of differentially expressed genes (DEGs) in total RNA (FC>1, $p<0.01$) (Fig 10B). Notable distinctions were observed in ribosome-bound mRNAs. When compared to the mock group, the HNP-Ribosome group displayed 192 down-regulated and 386 up-regulated genes. Conversely, the INP-Ribosome group showed 17 down-regulated and 446 up-regulated genes. Heatmaps visually represent the number of DEGs in each group (Fig 10C and 10D).

It's worth noting that the mRNA of several genes dependent on eIF4A1 translation is inhibited from loading onto ribosomes (Fig 10E). We selected Myc, cyclin B1, and cyclin A2 for

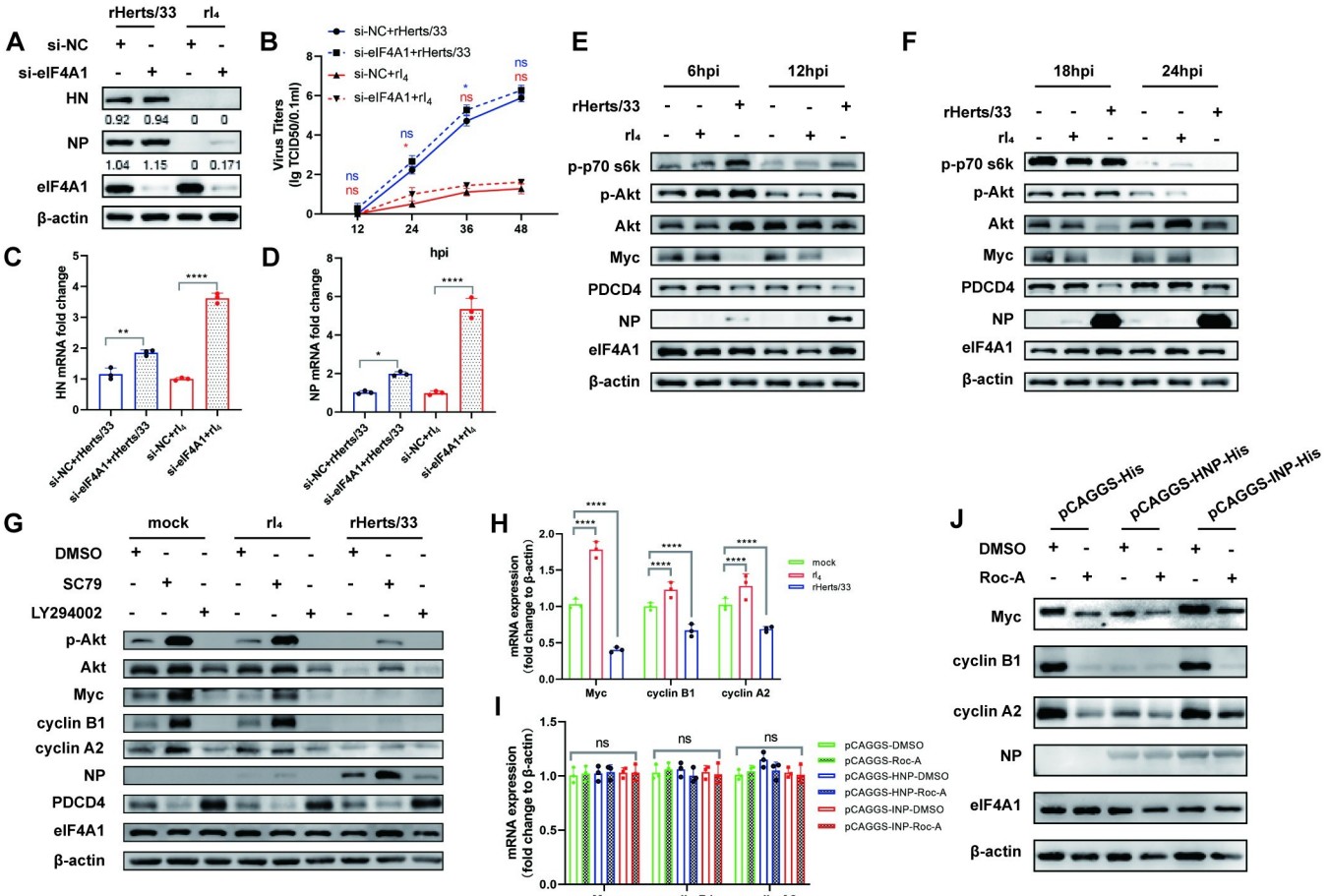

**Fig 9. HNP promotes NDV replication by inhibiting eIF4A1/Myc axis.** (A-D) HeLa cells were transfected with either si-NC or si-eIF4A1 for 24 h, followed by infection with rHerts/33 or rI4 at 1MOI. (A) Cells were harvested at 24 hpi to assess viral protein expression. (B) Cell culture supernatants were collected at 12-hour intervals post-infection until 48 hours, and the virus growth curve was determined by measuring TCID$_{50}$ values. (C-D) RNA was extracted at 24 hpi to examine the expression of virus-related mRNA. (E-F) HeLa cells, infected with either rHerts/33 or rI4, were harvested at 6, 12, 18, and 24 hpi for Western Blot analysis using specific antibodies. (G) HeLa cells were pre-treated with the Akt activator SC79 (5 μM) or the Akt inhibitor LY294002 (20 μM) for 2 hours, followed by infection with rHerts/33 or rI$_4$. Western Blot analysis was performed at 24 hpi using relevant antibodies. Equal volume DMSO treatment was served as a control. (H) HeLa cells were infected with rI4 and rHerts/33, and were harvested at 24 hpi to assess the mRNA expression of Myc, cyclin B1, and cyclin A2 in each group. The mRNA expression of target genes were normalized to β-actin mRNA and presented as fold induction. (I-J) HeLa cells were transfected with pCAGGS empty vector, pCAGGS-HNP-His, or pCAGGS-INP-His for 24 hours, followed by a 1-hour treatment with either DMSO or the eIF4A1 inhibitor Roc-A (3nM). In (I), the mRNA expression levels of target genes were measured and normalized to β-actin mRNA, presented as fold induction. In (J), Western Blot analysis was performed using relevant antibodies. Results are shown as the mean ± SD of three independent experiments. *$p < 0.05$ (considered as a significant difference), **$P < 0.01$, ***$P < 0.001$, ****$P < 0.0001$, ns (not significant) ≥ 0.05 (no significant difference).

validation, and the results were consistent with the RNC-Seq findings (Fig 10F–10H). Additionally, we observed that both HNP and INP could bind to the translationally active region within the cells (Fig 10I). Therefore, our results suggest that HNP, through its interaction with the translationally active region within cells, regulates the translation of cellular mRNA, and the inhibition of eIF4A1-related translation may be one of the mechanisms underlying this phenomenon.

## Discussion

As a highly virulent pathogen capable of infecting over 240 species of birds, NDV was initially identified in the 1950s for its selective infection of tumor cells [42]. Although various

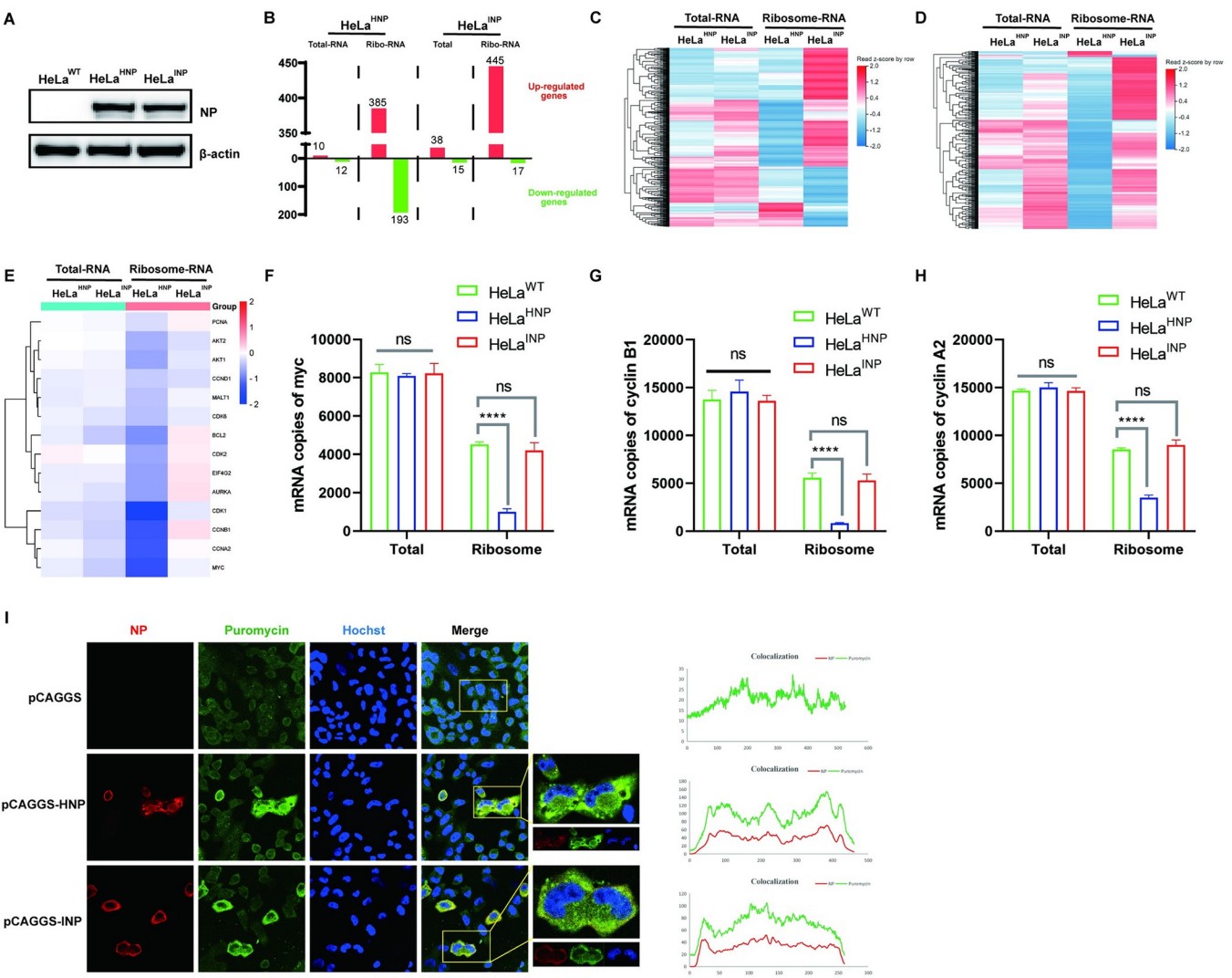

**Fig 10. HNP inhibits the eIF4A1-dependent translation.** (A) A lentiviral packaging system was used to express NDV-NP protein in HeLa cells, and NP expression was detected by Western blotting. (B) The statistical plot of the number of differentially expressed genes (DEGs) in each group (fc>1, $p$<0.01). Heat map of (C) HNP DEGs and (D) INP DEGs (fc>1, $p$<0.01). (E) Heatmap depicting the enrichment of genes dependent on eIF4A1-mediated translation in the ribosome. (F-H) Quantitative RT-PCR to detect the expression of cyclin A2 (F), Myc (G) and cyclin B1 (H). (I) Co-localization of NP protein with puromycin. Cells were transfected with pCAGGS-HNP-His, pCAGGS-INP-His, and control vector, followed by incubation with 208μM emetine at 37°C for 15 min to "freeze" the extended ribosome on the mRNA. This was followed by co-incubation with 10 μg/ml of puromycin (PMY, a tyr-tRNA mimetic) at 37°C for 5 min, and expression of NP and puromycin was observed by confocal microscopy using anti-His and anti-puromycin antibodies. Results are shown as the mean ± SD of three independent experiments. ****means $p$<0.0001.

lentogenic or mesogenic strains have been widely used to investigate oncolytic vaccines, their anti-tumor ability is relatively weak compared to velogenic viruses, which have a more vital ability to replicate in multiple cycles. Therefore, screening for more potent oncolytic strains and investigating their mechanisms remains of great interest. By evaluating the infectivity of several genotypes of NDV on a range of tumor cells, we found that NDV with the 450th-L-NP phenotype exhibited excellent oncolytic activity. Furthermore, we identified the leucine residue at the 450th amino acid position of the NP protein (450th-L-NP) as a critical factor in NDV infection of tumor cells by affecting the translation efficiency of the viral mRNA. Moreover, we have noted that the NP protein of the highly oncolytic Herts/33 strain interacts

directly with eIF4A1 via its 366-489aa, consequently obstructing eIF4A1-mediated mRNA translation. These two functionalities of the NP protein may jointly contribute to the preferential translation of NDV's viral mRNA.

In pathogenesis-related research, we found that the NDV with the 450th-F-NP phenotype demonstrated a substantial reduction in $TCID_{50}$ value, ranging from $10^5$ to $10^6$-fold, compared to the NDV with the 450th-L-NP phenotype, across five different tumor cell lines (Fig 1A–1E). Additionally, the direct oncolytic capability of NDV was positively correlated with its replicative capacity within cells, consistent with previous research findings [4,43–45]. Previous studies have indicated that each viral protein of NDV contribute to its pathogenicity. The NP and F proteins play significant roles in cross-host transmission in pigeons and chickens [46]. The L protein is the most significant contributor of the three polymerase-related proteins to viral pathogenicity in birds, and there is synergy between the homologous NP, P, and L proteins [47,48]. Additionally, the F and HN proteins have been shown to play essential roles in the viral infection of macrophages [49,50]. Our results indicate that replacing only the NP protein enables the non-oncolytic genotype VII $I_4$ strain to achieve infectivity in tumor cells at the level of the highly oncolytic Herts/33 strain (Fig 2F–2G). This suggests that NP plays an essential role in NDV infection of tumor cells. Notably, no previous studies have reported such a significant difference in $TCID_{50}$ value, where substituting a single gene can affect a $10^4$ to $10^6$-fold change. Since the mutation from phenylalanine to leucine at the 450th position of the NP protein resulted in a $10^{4-5}$-fold increase in $TCID_{50}$ value for the non-oncolytic $I_4$, this site was defined as the major determinant of NDV's differential infectivity in tumor cells (Fig 3I–3N). All paramyxovirus NP proteins contain IDR structural domains that often play a broad role in protein interactions [51] and are an important strategy for RNA viruses to buffer the deleterious effects of mutations [52]. The amino acid residue at position 402 of the first IDR of the NP N-Tail domain has previously been reported to affect polymerase activity [41]. The amino acid at position 450, located between two IDR regions of the NP N-Tail domain, may play a significant role in regulating the function of the disordered region. Notably, by comparing the sequences of 890 NDV strains, we found that 240 of the 251 genotype VII NDV strains are phenylalanine at position 450 of the NP protein, whereas almost all other strains are leucine in this site (Fig 4B–4D). Since we have demonstrated the role of amino acid 450 of the NP protein in NDV infection of mammalian cells, including tumor cells (Fig 3I–3N), we propose that NDV strains carrying a phenylalanine residue at position 450 of NP protein are not suitable for oncolytic NDV vaccines and mammalian vector vaccines. Simultaneously, the occurrence of this biological phenotype on genotype VII NDV, which is a relatively late-emerging strain, has prompted us to contemplate the evolutionary trajectory of NDV.

In the part of our investigation delving into the mechanisms behind the varying viral replication abilities, we uncovered several fascinating new findings. Our data confirm that in minigenome assays, the 450th amino acid position of the NP protein affects the translation of mRNA transcribed by the viral polymerase. It is known that viruses manipulate the host translation machinery to translate their mRNA. For instance, influenza virus RNA polymerase and NS1 proteins interact with eIF4G to recruit eIF4F [20], and the NP protein of NDV can bind to eIF4E [12], which may contribute to the translation of NDV mRNA. However, the mechanism by which the virus uses the modified translation initiation complex to recruit ribosomes still needs to be fully understood. Additionally, it is known that paramyxovirus mRNA is capped and methylated by the L protein, and NDV mRNA lacks 2'-O-methylation compared to eukaryotic mRNA [53]. Since the cap structure of mRNA plays a crucial role in enhancing its translation efficiency and stability [54,55], 450th-L-NP protein may play a critical role in recruiting cell factors to recognize the cap structure of viral mRNA and subsequently facilitate its effective loading onto ribosomes. However, the mechanism by which the 450th-L-NP

protein of NDV plays a functional role in promoting the loading of viral mRNA onto ribosomes remains to be studied.

After observing the correlation between 450th-L-NP and the translation of polymerase-transcribed mRNA in the microgenome, we attempted to replicate this phenomenon under conditions of viral infection. During viral infection, the efficiency of viral mRNA loading onto ribosomes from the 450th-L-NP phenotype of NDV remains significantly higher than that of the 450aa-F-NP phenotype of NDV when cellular translation activity is typically suppressed. Conversely, the translation efficiency of host mRNAs, including Type I interferon and β-actin, is markedly lower in cells infected with the 450th-L-NP phenotype of NDV compared to those infected with the 450th-F-NP phenotype of NDV. Viruses can ensure the preferential translation of viral mRNA by inhibiting the export of cellular mRNA from the nucleus or by usurping the cap structure of cellular mRNA [56]. Our results confirm the phenomenon of preferential translation of viral mRNA after NDV infection, but the underlying mechanism remains to be explored in future studies. Although there have been reports indicating that viruses selectively recruit translation factors to translate viral mRNA, ensuring the production of viral proteins and inhibiting host innate defense mechanisms by reducing the protein synthesis capacity of infected cells [21], our data can only suggest that, in cases where virus infection leads to the shutdown of host cell translation, the viral mRNA of the 450th-L-NP phenotype of NDV still maintains a relatively high translation level. In $rI_4$-infected cells, the recognition of viral components by intracellular Pattern Recognition Receptors (PRRs) initiates a more intense antiviral innate immune response [57,58,59]. This response has a dual nature: it is distinguished by a relatively higher efficiency in cellular mRNA translation and the incorrect expression of viral proteins that counteract the host's innate immune system, such as the v protein and NP protein [60,61]. At this point, while we have observed that the 450th-L-NP promotes the preferential translation of viral mRNA under the physiological conditions of virus infection, a deeper understanding of the cellular molecular mechanisms requires further exploration from the perspectives of interacting proteins and ribosome sequencing.

Following this, mass spectrometry was employed to search for interacting proteins of HNP and INP. The result revealed a direct interaction between eIF4A1 and HNP, while no such interaction was observed with INP (Fig 8). Knocking down the expression of eIF4A1 resulted in a mild promotion of NDV replication. This demonstrates that NDV replication is not contingent on eIF4A1, implying a relatively uncomplicated structural arrangement in the 5' UTR region of NDV viral mRNA. Similar to observations in various paramyxoviruses like Parainfluenza virus 5 (PIV5) and VSV, NDV replication activates the Akt pathway [62]. Interestingly, the activation of Akt did not lead to an upregulation of Myc expression. The phenomenon results from both HNP inhibiting translation activity dependent on eIF4A1 and the concurrent downregulation of mRNA for relevant genes during NDV infection. The interaction of the NP protein with eIF4A1 leads to the suppression of translation in mRNAs containing complex 5' UTR structures. However, since the mutation at the 450th amino acid does not affect the interaction between HNP and eIF4A1, and the knockdown of eIF4A1 has limited enhancing effects on viral replication, this is only a secondary factor in promoting viral replication. This further confirms that eIF4A1 does not regulate the translation of all mRNAs but rather a subset with complex structures in their 5' UTR [19,63]. Furthermore, the ability of NP protein to bind to cellular translationally active regions may serve as additional evidence supporting the role of NP proteins in modulating the translation of cellular mRNA (Fig 10I). Since eIF4A1-mediated mRNA translation largely promotes cancer progression, we may have unveiled a novel oncolytic mechanism of NDV that entails the targeted suppression of highly active oncogene expression in tumor cells. And, the inability of the NP protein of genotype VII $I_4$ to mediate efficient translation of viral mRNA may occur not only in tumor cells, but also occur in all mammalian

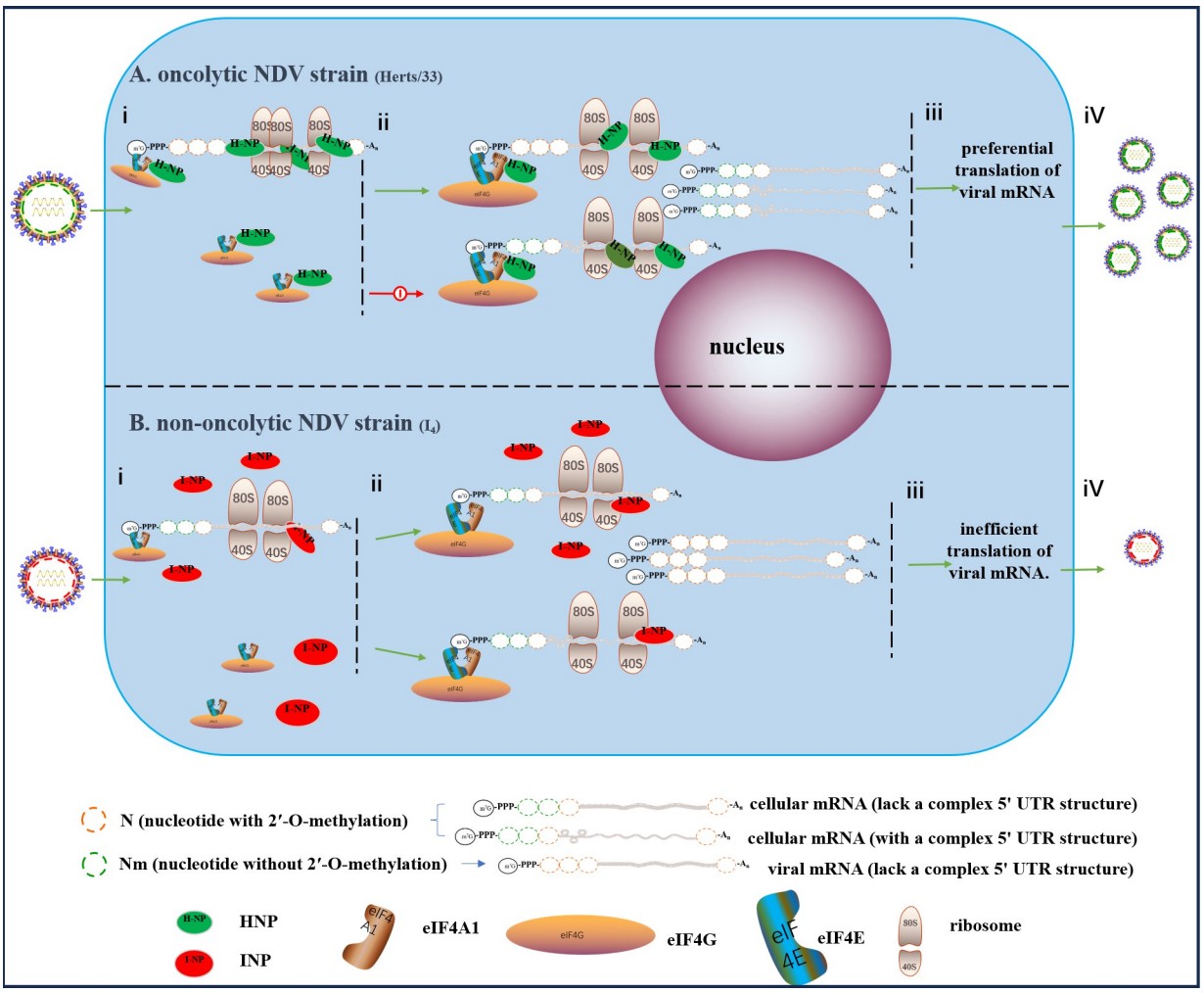

**Fig 11. Proposed model of the differential oncolytic capability mediated by NP Protein in NDV strains.** (A) Translation of oncolytic NDV (Herts/33) mRNA: i HNP(450th-L-NP) efficiently recruits ribosomes and interacts with eIF4A1 through its 366-489aa, ii The efficiency of viral mRNA lacking 2′-O-methylation modification loading onto ribosomes is higher. The translation of cellular mRNA with complex structures in the 5′UTR, dependent on eIF4A1, is inhibited. These two processes together resulting in low efficiency of cellular mRNA loading onto ribosomes and distribution in areas unrelated to ribosomes, iii Efficient translation of viral mRNA, iv Production of a large number of viral particles. (B) Translation of non-oncolytic NDV (I₄) mRNA: i INP(450th-F-NP) has weak binding ability with ribosomes and cannot interact with eIF4A1, ii Cellular mRNA containing 2'-O-methylation modification is efficiently loaded onto ribosomes. The translation of cellular mRNA with a complex structure in the 5'UTR, dependent on eIF4A1, is unaffected. Few viral mRNA is loaded onto ribosomes and more viral mRNA is distributed in regions unrelated to ribosomes, iii Extremely low efficiency of viral mRNA translation, iv Production of a small number of viral particles.

cells. This phenotype may be attributed to a significant divergence between translation-related proteins, including eIF4A1, in mammals compared to birds.

In summary, our research demonstrates that the NP protein is a key factor determining the infection of NDV on tumor cell lines. On one hand, its 450th amino acid affects the loading of viral mRNA onto ribosomes, while on the other hand, it can interact with eIF4A1 within the 366-489aa to inhibit eIF4A1-dependent cellular translation activity. These two mechanisms may work together to result in the preferential translation of viral mRNA (Fig 11). Our study provides novel insights into the function of the NP protein and suggests potential applications for oncolytic strain screening.

## Supporting information

**S1 Table. Construction strategy and primers for generating recombinant viruses.**
(DOCX)

**S2 Table. Construction strategy and primers for generating minigenome system-related plasmids.**
(DOCX)

**S3 Table. Primer sequences for the plasmids using the pCAGGS vector.**
(DOCX)

**S4 Table. Primers sequences used for the construction of plasmids in the BiFC experiment.**
(DOCX)

**S5 Table. Primer and probe sequences used in the qRT-PCR experiments.**
(DOCX)

**S6 Table. Primer sequences for the construction of NP protein prokaryotic expression plasmid.**
(DOCX)

**S7 Table. The sequence of siRNA used in this paper.**
(DOCX)

**S8 Table. Primers for generating HeLa cell lines with stable expression of the NP protein.**
(DOCX)

**S1 Fig. Infectivity of different genotypic strains on non-tumor mammalian cells.** Viral titers at 72hpi in (A) Vero, (B) sp2/0, (C) MDCK, (D) HcMEC/D3, (E) CHO, (F) 293T, (G) 16HBE, (H) $TCID_{50}$ value of ten additional genotype VII NDV strains in HeLa cells. Representative data, shown as the mean ± SD (n = 3), were analyzed with one-way ANOVA. ****, $P < 0.0001$.
(TIF)

**S2 Fig. No synergistic effect was found among the homologous NP, P, and L proteins.** (A) The cloning strategy schematic for simultaneous replacement of NP and P, NP, and L, or P and L genes between rHerts/33 and rI4. The virulence of the different recombinant viruses was determined by measuring the Intracerebral Pathogenicity Index (ICPI) in 1-day-old chickens. (B and C) $TCID_{50}$ value of the virulent strains after simultaneous replacement of NP and P, NP and L, or P and L genes. Representative data, shown as the means ± SDs (n = 3), were analyzed with two-way ANOVA. ****, $P < 0.0001$.
(TIF)

**S3 Fig. Prediction of disordered domains and infectivity assays of recombinant viruses.** (A and B) PONDR was used to predict the IDR of HNP (A) and INP (B) (http://www.pondr.com/ ). VL3 Predictor (Developed by P. Radivojac and A.K. Dunker) was used. Regions with a score greater than 0.5 are considered disordered regions. (C and D) PSIPRED was used to predict the IDR of HNP (A) and INP (B) (http://bioinf.cs.ucl.ac.uk/psipred/?disopred=1). Regions with a score greater than 0.5 are considered disordered regions. (E) Schematic diagram of the cloning strategy for exchanging the whole N-core and the first IDR of the N-tail domain between rHerts/33 and $rI_4$. The virulence of the different recombinant viruses was determined by measuring the ICPI in day-old chickens. (F, G, H, and I) $TCID_{50}$ value of recombinant strains after replacement of the whole N-core and the first IDR of N-tail domain. (J, K, L, M) $TCID_{50}$ values of mutant NDVs at 72hpi on several non-tumor cell lines. Representative data,

shown as the means ± SDs (n = 3), were analyzed with two-way ANOVA. ****, $P<0.0001$.
(TIF)

**S4 Fig. Differences occur during the viral protein translation.** (A) Expression of GFP was detected at 24 h in CEF cells after transfecting 0.5 μg or 1.5 μg minigenome with anti-GFP, anti-NP, and anti-β-actin. (B) HeLa cells were treated with 100μg/ml CHX for 1h and then infected with NDV (10MOI) at 37˚C for 0.5h. After that, cells were collected at 0h, 1h, 2h, 4h, and 8h.
(TIF)

**S5 Fig. Natural immunity and stress response are not the primary causes of differential infectivity.** HeLa cells were transfected with control siRNA or specific siRNA targeting RIG-I and PKR. After 36 h, cells were infected with NDV at 1 MOI or 10 MOI and collected at 24 hpi for analysis by immunoblotting with anti-RIG-I, anti-HN, anti-p-PKR, anti-PKR, or anti-β-actin antibodies.
(TIF)

**S6 Fig. Bioinformatics analysis of interacting proteins of HNP and INP.** Pathway analysis of the cellular proteins interacting with HNP (A) and INP (C) based on KEGG were performed using KEGG analysis tool in the Database for Annotation Visualization and Integrated Discovery (DAVID) (version 6.7). The Protein-Protein interaction networks of the cellular proteins interacting with HNP (B) and INP (D) was constructed using the Cytoscape software. Yellow nodes represent proteins specific to each interaction partner, while green nodes represent proteins that interact with both HNP and INP.
(TIF)

**S7 Fig. Schematic representation of RNA-Seq and RNC-Seq operations.** HeLa cells stably expressing different sources of NP and control HeLa cells were used to isolate ribosomes and enrich mRNA for multimeric fractions for RNC-Seq, and the other half were directly extracted for total RNA and enriched mRNA for RNA-Seq.
(TIF)

## Acknowledgments

We gratefully acknowledge Prof. Xiulong Xu (Yangzhou University, China) and Prof. Chan Ding (Shanghai Veterinary Research Institute, CAAS) for giving constructive suggestions. We appreciate Prof. Xiulong Xu (Yangzhou University, China) for generously providing the SW1736 cell line.

## Author Contributions

**Conceptualization:** Tianxing Liao, Yu Chen.

**Data curation:** Tianxing Liao, Lili Guo.

**Formal analysis:** Xiaolong Lu, Xiaowen Liu, Xiaoquan Wang.

**Funding acquisition:** Yu Chen, Haixu Xu, Zenglei Hu, Xiufan Liu.

**Investigation:** Tianxing Liao, Lili Guo.

**Methodology:** Tianxing Liao, Lili Guo, Tiansong Zhan, Haixu Xu.

**Project administration:** Xiufan Liu.

**Resources:** Xiaolong Lu, Min Gu, Xiaowen Liu, Xiaoquan Wang.

**Software:** Tianxing Liao, Zenglei Hu, Jiao Hu.

**Supervision:** Xiufan Liu.

**Validation:** Tianxing Liao, Yu Chen.

**Visualization:** Tianxing Liao, Shanshan Zhu.

**Writing – original draft:** Tianxing Liao, Yu Chen.

**Writing – review & editing:** Yu Chen, Shunlin Hu, Xiufan Liu.

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
