## [Decision Letter · Decision Letter 0]

30 Jun 2023

Dear Prof. Liu,

Thank you very much for submitting your manuscript "The 450th amino acid of NP protein determines the oncolytic activity of Newcastle Disease virus by regulating viral mRNA translation efficiency" for consideration at PLOS Pathogens. As with all papers reviewed by the journal, your manuscript was reviewed by members of the editorial board and by several independent reviewers. In light of the reviews (below this email), we would like to invite the resubmission of a significantly-revised version that takes into account the reviewers' comments.

In particular, please note the comments of Reviewer #1 on the need for additional mechanistic insight into your findings and the comments of Reviewer #2 regarding some of your experimental approaches. A revised manuscript must thoroughly address these critiques.

We cannot make any decision about publication until we have seen the revised manuscript and your response to the reviewers' comments. Your revised manuscript is also likely to be sent to reviewers for further evaluation.

Sincerely,

Michael D Robek

Academic Editor

PLOS Pathogens

Meike Dittmann

Section Editor

PLOS Pathogens

Kasturi Haldar

Editor-in-Chief

PLOS Pathogens

orcid.org/0000-0001-5065-158X

Michael Malim

Editor-in-Chief

PLOS Pathogens

orcid.org/0000-0002-7699-2064

Reviewer's Responses to Questions

**Part I - Summary**

Reviewer #1: Liao and colleagues investigated an unexpected observation that the replication of virulent NDV strains belonging to genotype VII is severely restricted in multiple human cancer cell lines. Productive NDV replication is highly specific to avian hosts, and it is generally believed that the major restriction factor in mammals is the inefficient inactivation of the interferon response by the viral protein V. This mediates the selective replication of NDVs in cancer cells with defects in anti-viral signaling making them promising oncolytic agents. In this case, however, the authors by making chimeric constructs between the NDV strain effectively replicating in cancer cell lines (Herts/33) and the representative of the genotype VII (I4) narrowed down the restriction determinant to amino-acid 450 in the nucleocapsid protein (NP), which is F in genotype VII (replication restricted) and L in other genotypes (replication-competent). Further analysis using polysome profiling and minigenome replication systems revealed that NP with L450 was associated with polysomes at a significantly higher level than F450, and accordingly, more GFP-coding mRNA generated by the minigenome system was found in the polysomal fraction with NP with L450. The authors also identified that amino-acids 122-366 of NPs contain the domain required for the association of NP with polysomal fraction. Altogether, the presented data show that NPs with L450 are competent in promoting translation of virus-generated mRNAs in mammalian cells, while those with F450 are not.

The observation of a specific NP-mediated restriction of genotype VII replication in mammalian cells is interesting, but the mechanistic insights are insufficient. The role of NDV NPs in promoting virus-specific mRNA translation has been reported before (PMID: 32603377) and was attributed to the interaction of NP with the translation initiation factor eIF4E. Also, the data on the presumable specific inhibition of translation of IFN-coding mRNAs by the NP with L450 (from replication-competent viruses) are overinterpreted:

Fig. 7. The results reflect the differences of the I4 and Hert 33 replication, leading to a stronger general suppression of host mRNA translation, not a preferential inhibition of IFN mRNA translation in the latter case

Fig. 7E qPCR data only show the amount of NP-coding mRNA, these data can not be extrapolated to all virus-generated mRNAs and should be described accordingly (line 600)

Reviewer #2: In this manuscript, the authors discovered that the 450th amino acid of the NP protein is critical for the oncolytic activity of NDV strains. Their subsequent experiments revealed that amino acid differences in the 450th position of NP affected its binding efficiency with mammalian polysomes. Their finding is highly suggestive, as it answers a long-standing question about why the oncolytic activity of NDV appears to be unrelated to its virulence. The results explain this phenomenon from the perspective of host selective translation and have great significance in guiding research on oncolytic viruses. There is a great deal of work here. The authors are certainly to be commended for their effort in telling a complete story. They present a fairly strong case for the conclusions listed above. However, there are a number of significant issues that should be addressed in a revised version, as well as numerous minor points.

**Part II – Major Issues: Key Experiments Required for Acceptance**

Reviewer #1: The authors should investigate the mechanism of the defectiveness of NPs with F450 in activating viral mRNA translation in mammalian cells. The previous work (PMID: 32603377) may provide some useful guidence

Reviewer #2: Line148-152: The cell infection of lentogenic NDV strains, like LaSota, requires extra addition of trypsin. It was not mentioned in the Methods. The concentration of trypsin in the medium is also a major factor affecting the infection efficiency of NDV strains in cells.

Line 236: there are some problems in the methods of virus binding assay. It is recommended that cells were incubated with NDV at 4℃ to prevent virus entry in the process of absorption.

Some of the author’s conclusions appear to lack rigor. For instance, the study only examined a limited number of genotype VII strains, yet the conclusion drawn was that genotype VII NDVs are not oncolytic. It is unclear whether this trait is specific to a group of strains or several genotypes. Furthermore, there is a possibility of genotype VII strains containing the F450L substitution in NP, which could provide oncolytic activity to these strains. The authors should carefully analyze the outcomes of their experiments and rigorously draw conclusions.

The authors constructed a great number of recombinant viruses and plasmids, and their labelling lacks clarity, resulting the difficulty of understanding. Therefore, a more unambiguous and concise naming system is suggested for improved comprehension. Additionally, some strains have associated schematic diagrams, others do not. For instance, the rl-NPPLH and rH-NPPLL is the main research object in line 384-392, but their schematic diagrams were not displayed in Figure 1A and the background was not clear.

Figure 3J-M, it is recommended to detect and compare TCID50 titers of these strains on CEF.

Line 472-487: it is suggested that the amino acid sequences of NP gene from classical oncolytic strains should be specially taken for alignment.

Line 499-500: According to the flow cytometry results in Figure 5A, the adsorption rate of rl4 is higher than that of rHerts/33. The flow cytometry assay should be repeated for no less than three times and the data of virus adsorption should be compared statistically.

Line 505-517: Based on the WB assay shown in Figure 5H, it appears that the HNP protein decreases due to CHX treatment. The author should repeat this experiment no less than three times, scan the gray scale, and perform statistical analysis. In my opinion, the results here suggest that the functional difference between HNP and INP may lie in translation, since CHX is an inhibitor of translation elongation.

Figure 6A: RNase digestion is recommended here to determine whether NP or extra RNA is responsible for the difference in polysome amounts.

In the RNC-seq assay, why did the authors use HNP and INP rather than 450aa mutants?

In all of the figures' bar graphs, the distinctions between the colors representing various samples are not clear. Thus, it is recommended to use colors that have more noticeable differences to present the data.

Finally, the discussion is quite lengthy and aimless. It is advisable to restructure the language by clarifying the topic and removing any redundant information.

**Part III – Minor Issues: Editorial and Data Presentation Modifications**

Reviewer #1: Lines 50-52. P and L are not structural proteins

Line 58. Replace “cellular activity” with cellular viability

Line 74. Spell out MV (measles virus)?

Line 86. Translation of viral mRNAs, not proteins. Information in RNA is being translated into protein.

Line 89. Spell MeV (measles virus)?

Line 351. Fix genotype number

Line 362. Replace “cellular activity” with cellular viability

Line 420. Remove “typically”

Lines 444-445. These are not mutations, these are differences in amino-acid composition of the NP proteins of the two strains

Line 625. Change “tir” to tyr (tyrosine)

Lines 647-648. It is not clear what did the authors mean by “both HNP... promote translational activity by targeting translationally active regions, but HNP can selectively regulate cellular mRNA translation”

Reviewer #2: L23: “the remaining genotypes” is supposed to be “the other genotype” or “the genotypes of rest”

L56-57: Does the sentence mean “there is no correlation between the oncolytic properties of NDV and its virulence”?

Line320: define “RNC-Seq” when the abbreviation was first used.

Line351: genotype what?

Figure1A-F: lack detection time point.

Line387: the word “compatible” is not correct here.

Line456: The meaning of this sentence is not clear.

Figure 6E: “Ploysomes” should be “Polysomes”.

In the whole manuscript, all the log10 should be written as lg.

The author used the words minigenome and microgenome interchangeably. Please use the same name.

PLOS authors have the option to publish the peer review history of their article (what does this mean?). If published, this will include your full peer review and any attached files.

Reviewer #1: No

Reviewer #2: No
---

## [Decision Letter · Decision Letter 1]

24 Oct 2023

Dear Prof. Liu,

Thank you very much for submitting your manuscript "NP protein determines the oncolytic activity of Newcastle Disease virus by regulating viral mRNA translation efficiency" for consideration at PLOS Pathogens. As with all papers reviewed by the journal, your manuscript was reviewed by members of the editorial board and by several independent reviewers. The reviewers appreciated the attention to an important topic. Based on the reviews, we are likely to accept this manuscript for publication, providing that you modify the manuscript according to the review recommendations.

Sincerely,

Meike Dittmann, Ph.D.

Section Editor

PLOS Pathogens

Meike Dittmann

Section Editor

PLOS Pathogens

Kasturi Haldar

Editor-in-Chief

PLOS Pathogens

orcid.org/0000-0001-5065-158X

Michael Malim

Editor-in-Chief

PLOS Pathogens

orcid.org/0000-0002-7699-2064

Reviewer Comments (if any, and for reference):

Reviewer's Responses to Questions

**Part I - Summary**

Reviewer #1: The authors did a great job improving the manuscript and I am willing to support its publication provided they will incorporate the figure they placed in the reponse to reviewers letter in the paper. It is not a good practice to provide data for reviewers only, if the authors think they are relevant to the work, they should be available to all readers of the paper.

Reviewer #2: The authors discovered that the 450th amino acid of the NP protein is crucial for the oncolytic activity of NDV strains in this manuscript. Following many strong tests, they revealed that amino acid variations in the 450th position of NP influenced its binding efficiency with mammalian polysomes. There is a lot of work to be done here,and their discovery perfect answer that why NDV's oncolytic function appears to be unrelated to its pathogenicity.

Following the initial round of revision,the corresponding authors fully responded to the two reviewers' comments and rectified the faults in the manuscript to make it easier to understanding.

Here i have two suggestions

1. Please recheck the manuscript again to avoid the language mistake.For example line 26 “450th-F-NP exhibits weaker functionality in this process.”may not suitable to be a complete sentence in the abstract.

2. The conclusion can be substantially supported by the experimental findings, and the viewpoint is also highly original. However, there is lack of an easy-to-understand pattern diagram to help readers rapidly understand the article's main themes. Please make the manuscript more legible by add a schematic plot, if possible.

**Part II – Major Issues: Key Experiments Required for Acceptance**

Reviewer #1: (No Response)

Reviewer #2: (No Response)

**Part III – Minor Issues: Editorial and Data Presentation Modifications**

Reviewer #1: (No Response)

Reviewer #2: (No Response)

PLOS authors have the option to publish the peer review history of their article (what does this mean?). If published, this will include your full peer review and any attached files.

Reviewer #1: No

Reviewer #2: No

Figure Files:

Data Requirements:

Reproducibility:

References:

---

## [Editor Report · Decision Letter 2]

9 Jan 2024

Dear Prof. Liu,

Thank you very much for submitting your manuscript "The NP protein of Newcastle disease virus dictates its oncolytic activity by regulating viral mRNA translation efficiency" for consideration at PLOS Pathogens. As with all papers reviewed by the journal, your manuscript was reviewed by members of the editorial board, by several independent reviewers, and by the PLOS Dual-Use Research of Concern (DURC) committee. Thank you for your patience in this process. Based on the reviews, we are likely to accept this manuscript for publication, providing that you modify the manuscript according to the review and DURC recommendations.

Sincerely,

Meike Dittmann, Ph.D.

Section Editor

PLOS Pathogens

Meike Dittmann

Section Editor

PLOS Pathogens

Kasturi Haldar

Editor-in-Chief

PLOS Pathogens

orcid.org/0000-0001-5065-158X

Michael Malim

Editor-in-Chief

PLOS Pathogens

orcid.org/0000-0002-7699-2064

NDV is a US Federal Select Agent, and so your excellent work is considered potential DURC. The committee notes that the paper includes NDV variants that are (as you show) virulent, but also notes that the experiments are focused on tumor killing and not on infectivity or virulence of the NDV variants. The main concern is that the paper currently lacks any information about biosafety and does not describe institutional oversight and approval of the research. Therefore, please add a section to Materials and Methods that describes procedures and facilities for biosafety, in detail. In addition, please name the institutional committee(s) responsible for approval and oversight of the research.

Reviewer Comments (if any, and for reference):

Figure Files:

Data Requirements:

Reproducibility:

References:

---

## [Editor Report · Decision Letter 3]

5 Feb 2024

Dear Prof. Liu,

We are pleased to inform you that your manuscript 'The NP protein of Newcastle disease virus dictates its oncolytic activity by regulating viral mRNA translation efficiency' has been provisionally accepted for publication in PLOS Pathogens.

Best regards,

Meike Dittmann, Ph.D.

Section Editor

PLOS Pathogens

Meike Dittmann

Section Editor

PLOS Pathogens

Michael Malim

Editor-in-Chief

PLOS Pathogens

orcid.org/0000-0002-7699-2064
---

## [Editor Report · Acceptance letter]

15 Feb 2024

Dear Prof. Liu,

We are delighted to inform you that your manuscript, "The NP protein of Newcastle disease virus dictates its oncolytic activity by regulating viral mRNA translation efficiency," has been formally accepted for publication in PLOS Pathogens.

Best regards,

Michael Malim

Editor-in-Chief

PLOS Pathogens

orcid.org/0000-0002-7699-2064